# Enhancement of trans-cleavage activity of Cas12a with engineered crRNA enables amplified nucleic acid detection

Long T. Nguyen[1], Brianna M. Smith[1] & Piyush K. Jain[1,2 ✉]

The CRISPR-Cas12a RNA-guided complexes have tremendous potential for nucleic acid detection but are limited to the picomolar detection limit without an amplification step. Here, we develop a platform with engineered crRNAs and optimized conditions that enabled us to detect various clinically relevant nucleic acid targets with higher sensitivity, achieving a limit of detection in the femtomolar range without any target pre-amplification step. By extending the 3'- or 5'-ends of the crRNA with different lengths of ssDNA, ssRNA, and phosphorothioate ssDNA, we discover a self-catalytic behavior and an augmented rate of LbCas12a-mediated collateral cleavage activity as high as 3.5-fold compared to the wild-type crRNA and with significant improvement in specificity for target recognition. Particularly, the 7-mer DNA extension to crRNA is determined to be universal and spacer-independent for enhancing the sensitivity and specificity of LbCas12a-mediated nucleic acid detection. We perform a detailed characterization of our engineered ENHANCE system with various crRNA modifications, target types, reporters, and divalent cations. With isothermal amplification of SARS-CoV-2 RNA using RT-LAMP, the modified crRNAs are incorporated in a paper-based lateral flow assay that can detect the target with up to 23-fold higher sensitivity within 40–60 min.

[1] Department of Chemical Engineering, University of Florida, 1006 Center Drive, Gainesville, FL 32611, USA. [2] UF Health Cancer Center, University of Florida, 2033 Mowry Rd., CGRC 463, Gainesville, FL 32608, USA. ✉email: jainp@ufl.edu

Class 2 CRISPR-Cas (Clustered Regularly Interspaced Short Palindromic Repeats/CRISPR-associated proteins) systems, such as Cas12a (previously referred as Cpf1, subtype V-A) and Cas13a (previously referred C2c2, subtype VI), are capable of nonspecific cleavage of ssDNA (single-stranded DNA) and RNA, respectively, in addition to successful gene editing[1–3]. This attribute, known as trans-cleavage, is only activated once bound to an activator (ssDNA or dsDNA) that has complementary base-pairing to the guide crRNA. When combined with a FRET-based reporter, a fluorophore connected to a quencher via a short oligonucleotide sequence, the presence of the target activator can be confirmed. This phenomenon has been efficiently harnessed by SHERLOCK (Specific High-sensitivity Enzymatic Reporter unLOCKing) and DETECTR (DNA Endonuclease Targeted CRISPR Trans Reporter) to reliably detect nucleic acids[1,4–8].

Research studies have reported that an extended secondary DNA on the guide crRNA for Cas12a or a hairpin RNA structure added to the sgRNA for Cas9 increases the efficiency and specificity of gene editing[9,10]. Besides, chemically modified Cas12a guided-RNA has also been shown to facilitate improved gene correction in mammalian cells through both viral and non-viral methods compared to the wild-type guide RNA[11]. Though these modifications are employed to utilize the cis-cleavage aspect of the CRISPR-Cas systems, the effects of such alterations on the trans-cleavage remain unknown.

Based on the crystal structure of LbCas12a-crRNA-dsDNA (PDB ID: 5XUS)[12], we reasoned that crRNA extensions can influence the trans-cleavage activity by either activating or inhibiting the catalytic efficiency of Cas12a, allowing us to better understand crRNA design with tunable trans-cleavage activity. We speculated that nucleic acid extensions on the crRNA can potentially change its nature of binding and subsequently alter this collateral cleavage due to conformational changes of the Cas12a dynamic endonuclease domain.

In this study, we engineer crRNAs with high sensitivity and specificity for detecting nucleic acids using CRISPR-Cas12a. We optimized the previously developed CRISPR-based detection assays[1,5,6] and combined them with our engineered crRNA with DNA extensions to create a CRISPR-ENHANCE (ENHanced Analysis of Nucleic acids with CrRNA Extensions) technology or referred here as ENHANCE. Next, we apply the ENHANCE towards enhanced detection of a variety of clinically relevant targets including PCA3, HIV, HCV, and SARS-CoV-2. Notably, by combining an isothermal amplification step, this system shows improved detection of SARS-CoV-2 genomic RNA using a fluorescence-based and a paper-based lateral flow assay compared to wild-type CRISPR-Cas12a system.

## Results

**crRNA extensions affect Cas12a trans-cleavage activity.** We placed ssDNA, ssRNA, and phosphorothioate ssDNA extensions of various lengths ranging from 7 to 31 nucleotides on either the 3′- or 5′-ends of the crRNA targeting GFP (green fluorescent protein), referred to here as crGFP (Fig. 1b–h). To measure the collateral or trans-cleavage activity of Cas12a, we employed a FRET-based reporter used in DETECTR[1], composed of a fluorophore (FAM) and a quencher (3IABkFQ) connected by a 5-nucleotide sequence (TTATT), which displays increased fluorescence upon cleavage. Consistent with the previous literature[13], when using wild-type crRNAs, we observed that the LbCas12a exhibited higher trans-cleavage activity than the AsCas12a or the FnCas12a, and therefore, we designed various modified crRNAs compatible with LbCas12a. Using the same reporters, we discovered that ssDNA and ssRNA extensions on the 3′-end of crGFP markedly enhanced the trans-cleavage ability of target-activated LbCas12a. Comparing the two types, the ssDNA extensions demonstrated higher activity than the corresponding ssRNA (Fig. 1b–d, f and Supplementary Figs. 1–5). On the other hand, the phosphorothioate ssDNA extensions at the 3′-end or 5′-end displayed minimal or no activity, showing decreased fluorescence intensity as modification length increased (Fig. 1e, h and Supplementary Figs. 1–5). This observation suggests that further extending the crRNA with 13-mer phosphorothioate ssDNA and beyond significantly inhibits LbCas12a trans-cleavage activity. The finding corroborated B. Li and colleagues that phosphorothioate ssDNA may prevent crRNA-Cas12a-DNA complex formation[14].

Notably, the 3′-DNA with 7-mer extensions on the crGFP referred to as crGFP + 3′DNA7, yielded the highest fluorescence signal compared to other modifications, measuring approximately 3.5-fold higher intensity than the wild-type crGFP (Fig. 1c, Supplementary Figs. 1a, 2a). By investigating the conformational changes from the crystal structure of the binary LbCas12a:crRNA complex[12,15,16], we observed that the 3′-end modifications on crRNA are proximal to the RuvC region of the LbCas12a. This supports our observation that the 3′-end extensions lead to higher trans-cleavage activity than the 5′-end. We hypothesized that the reporter composition itself may affect the LbCas12a collateral cleavage activity. Therefore, we incorporated and tested various nucleotides (GC and TA-rich) and fluorophores (FAM, HEX, and Cy5) within the reporter. Consistent with our hypothesis, we observed that the LbCas12a achieved maximal trans-cleavage activity with FAM or HEX and TA-rich reporter (Fig. 2a and Supplementary Figs. 1–5).

**DNA-extended crRNA enhances the rate of trans-cleavage.** We speculated that once an R-loop is formed between crRNA and dsDNA or ssDNA activator, the LbCas12a executes a partial trans-cleavage of the 3′-end of crRNA, leaving an overhang. These remaining extensions may further expand the nuclease domain in the LbCas12a, resulting in conformational changes and allowing more access for nonspecific ssDNA cleavage. To confirm our hypothesis, we attached different fluorophores, or fluorophore-quencher pairs separated by DNA linkers, to either the 3′-end or 5′-end of the crGFP with 7-mer DNA extensions and analyzed by denaturing gel electrophoresis. Surprisingly, we discovered that the 3′-end of the crRNA is processed by LbCas12a only in the presence of an activator while the 5′-end is cleaved by LbCas12a even in the absence of the activator (Fig. 2b, c and Supplementary Figs. 6, 7).

To further understand the LbCas12a enhanced enzymatic activity, we performed a Michaelis-Menten kinetic study on the wild-type crGFP and the crGFP + 3′DNA7 and observed that the ratio Kcat/Km was 3.2-fold higher for crGFP + 3′DNA7 than the unmodified crGFP (Fig. 2d, e). The time-dependent gel electrophoresis analysis of nonspecific cleavage of ssDNA M13mp18 phage (~7 kb) reconfirmed the fluorophore-quencher-based reporter assay results (Fig. 2f).

Based on our findings that the trans-cleavage activity is drastically improved by 7-mer ssDNA extensions to the 3′-end of crGFP, we questioned if the binding of crRNA with LbCas12a itself is influenced by such modifications. A biolayer interferometry binding kinetic assay revealed that the dissociation constant, $K_d$, between the binary complex LbCas12a:crRNA and LbCas12a:crRNA + 3′DNA7 are comparable within a low nM scale (Fig. 2g and Supplementary Fig. 8). These binding results suggest that the 3′DNA7 modification on crRNA does not affect the binary complex formation between the LbCas12a and the crRNA.

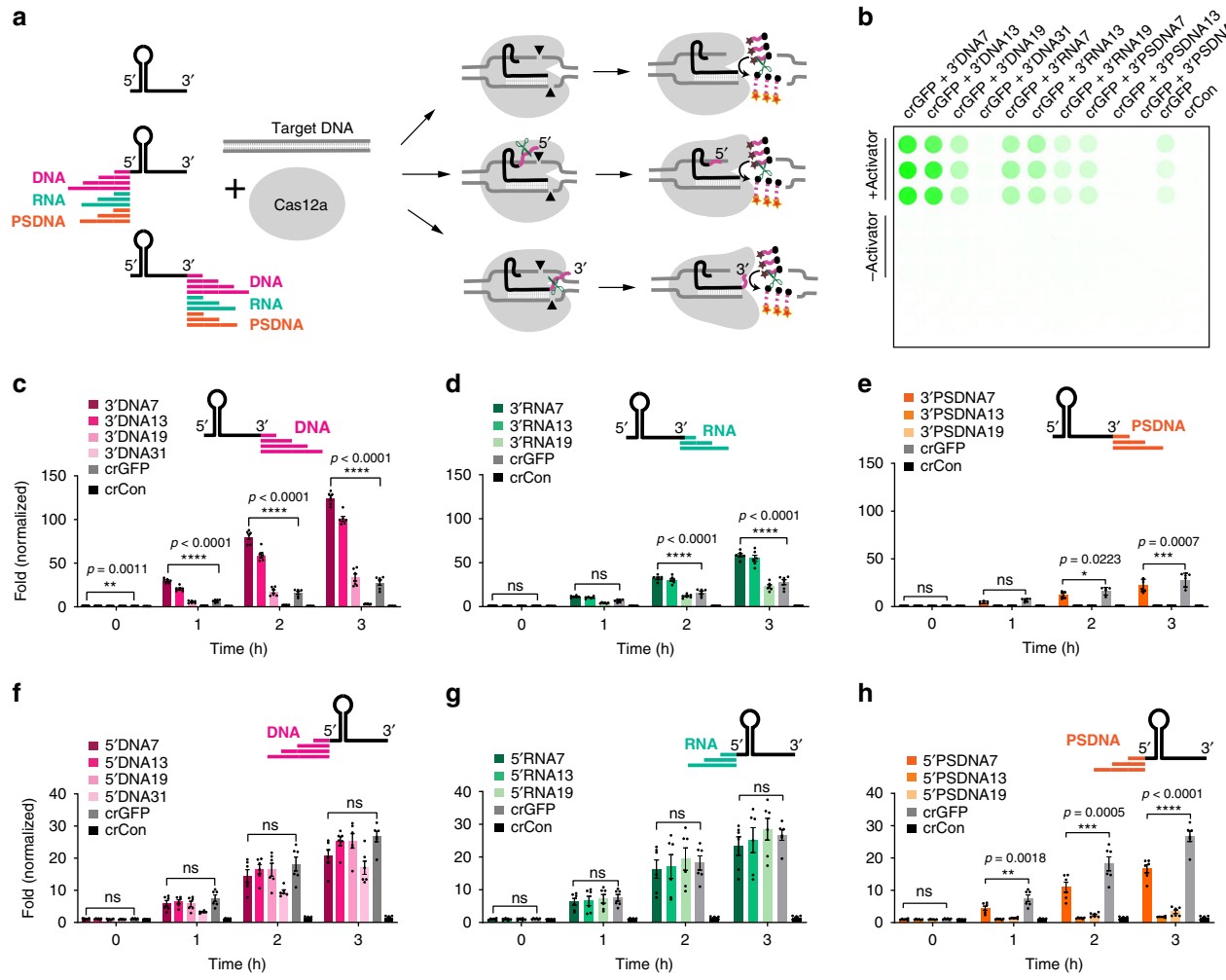

**Fig. 1 The trans-cleavage activity of LbCas12a with modified crRNA via fluorescence-quencher-based reporter assay with TA rich fluorophore-quencher systems tested.** For other fluorophore-quencher systems, see Supplementary Figs. 1–4. **a** Schematic diagram of cleavage of Cas12a with wild-type and modified crRNAs. The crRNA is extended on either the 3′- or 5′-ends with ssDNA, ssRNA, or phosphorothioate ssDNA. **b** A representation of a fluorescence-quencher-based trans-cleavage reporter assay image taken by GE Amersham Typhoon. **c**, **d**, and **e** 3′-end ssDNA, ssRNA, and phosphorothioate ssDNA extensions of crRNA, respectively. **f**, **g**, and **h** 5′-end ssDNA, ssRNA, and phosphorothioate ssDNA extensions of crRNA, respectively. The fold in fluorescence was normalized by taking the ratio of background-corrected fluorescence signal of a sample with the activator to the corresponding sample without activator. For **c**–**h**, error bars represent mean ± SEM, where $n = 6$ replicates (three technical replicates examined over two independent experiments). Statistical analysis was performed using a two-way ANOVA test with Dunnett's multiple comparison test, where ns = not significant with $p > 0.05$, and the asterisks (*, **, ***, ****) denote significant differences with $p$ values listed above. Source data are available in the Source Data file.

**LbCas12a shows the highest activity with TA-rich extended crRNAs.** By placing the fluorophore FAM on the 5′-end and a 7-mer DNA extension on the 3′-end of the crGFP, we learned that the first Uracil on the 5′-end of the crGFP gets trimmed by LbCas12a in the absence of an activator, which corroborated previous studies reported for FnCas12a[12] (Fig. 2b). We hypothesize that the 5′-end modifications are eliminated and converted back to the wild-type crRNA before complexing with the activator. This finding reinforces our previous observation that the 5′-extended crRNA has similar collateral cleavage activity as the wild-type crRNA.

Fascinated by LbCas12a pre-crRNA processing as previously described[17] and from our observations, we investigated how extensions of the mature crRNA would influence the trans-cleavage activity compared to the corresponding extended pre-crRNA. We discovered that the modified pre-crRNA and modified mature crRNA (tru-crRNA) exhibited comparable trans-cleavage efficiency (Fig. 3a). Furthermore, when a dsDNA

or an ssDNA activator was present, the 3′-end and 5′-end DNA-extended crRNA were cleaved (Fig. 2b, c and Supplementary Figs. 6, 7). These results led us to question whether the trans-cleavage activity is dependent on the sequence of ssDNA extensions on the 3′-end of the crRNA. To test this, we altered the nucleotide content of the extended regions of the crGFP. It turned out that the crGFP with TA-rich extensions carried out significantly more collateral cleavage than those with GC-rich regions (Fig. 3b and Supplementary Fig. 9).

Fascinated by the enhanced trans-cleavage activity of the crRNA + 3′DNA7, we sought to further explore the possibility of this modified crRNA by carrying out experiments on non-fully phosphorothioate of the crGFP + 3′DNA7 with 1 to 6 PS substitutions starting from the 3′-end of the extension inwards. We were interested in understanding if the trans-cleavage activity of LbCas12a could be enhanced further by protecting the DNA extension with phosphorothioate modifications. Interestingly, fluorescence-based reporter assays showed that the LbCas12a

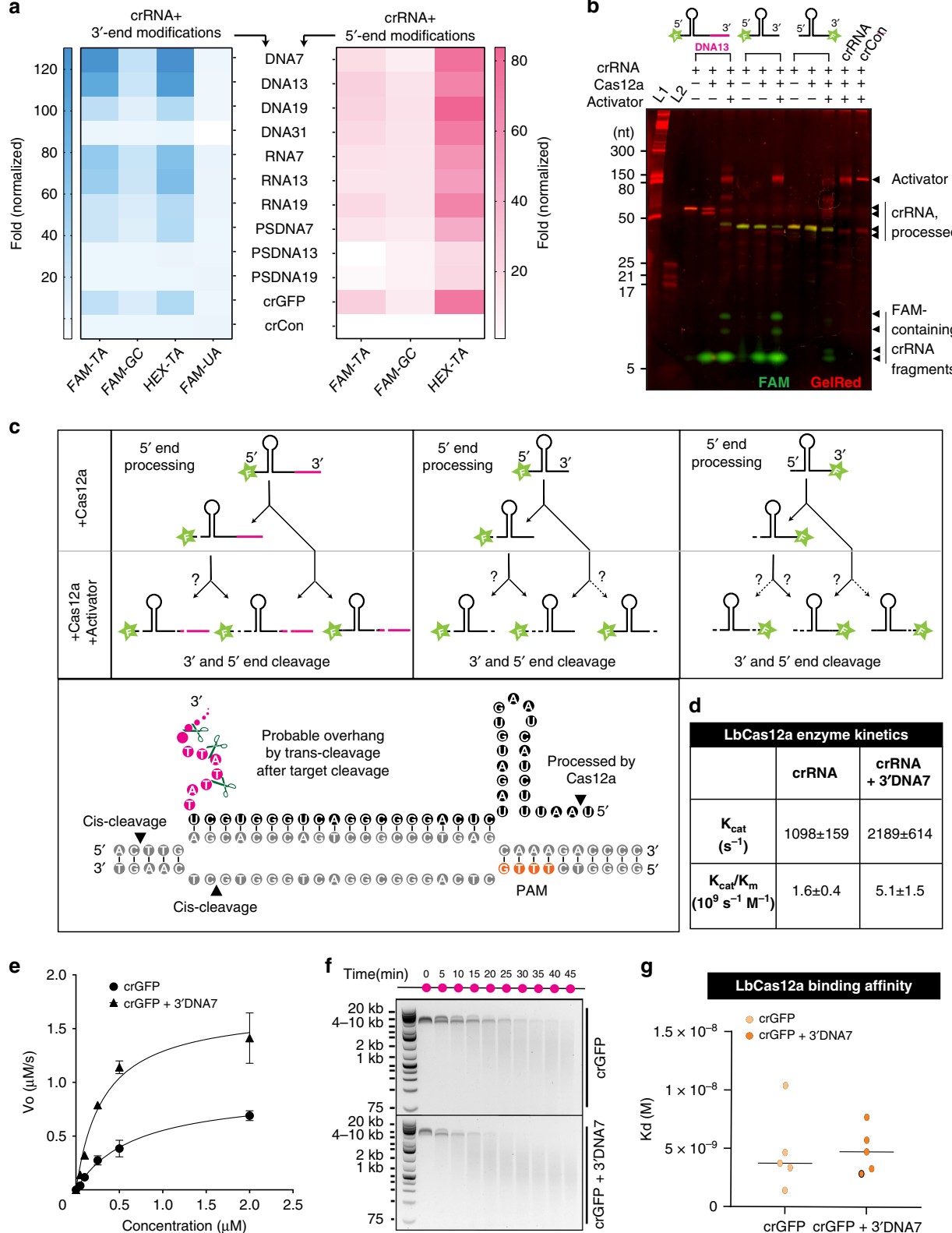

trans-cleavage activity decreased as more phosphorothioate modifications were added to the extension, with the non-phosphorothioated crRNA + 3′DNA7 exhibiting the highest fluorescence signal (Fig. 3c, d and Supplementary Fig. 10).

While 7-mer ssDNA extensions on the 3′-end of crRNA increase trans-cleavage activity with LbCas12a, we questioned if this is consistent across other orthologs of Cas12a. To investigate

further, we carried out an in vitro cis-cleavage and trans-cleavage assay of AsCas12a and FnCas12a with an extended crGFP compared to a wild-type crGFP (Fig. 3e and Supplementary Fig. 11). Interestingly, the crGFP + 3′DNA7 showed similar results with FnCas12a; however, it exhibited an opposite effect with AsCas12a. On the other hand, the cis-cleavage activity was found to be comparable between the crGFP and crGFP + 3′DNA7

**Fig. 2 Effect of reporter sequences on trans-cleavage activity, the proposed mechanism of crRNA end processing, enzyme kinetics, and binding affinity of LbCas12a with modified crGFP. a** Effect of different types of fluorophore-quencher systems on trans-cleavage activity with various modifications of crRNA. **b** Interactions of fluorescently labeled crRNAs with LbCas12a and dsDNA activator, characterized by PAGE analysis. In the absence of the activator, the modified crRNA (pre-crRNA) is trimmed by LbCas12a on its 5′-end (the first Uracil is cleaved, so-called truncated-crRNA or tru-crRNA). In the presence of the activator, the crRNA extensions are further trimmed, possibly leaving a 3′overhang. **c** Schematic diagram of putative processing of crRNA cleavage sites in the presence and absence of activator GFP. **d** Enzyme kinetic data of LbCas12a with crGFP vs. crGFP + 3′DNA7. **e** Michaelis-Menten kinetic study of the wild-type crGFP vs. crGFP + 3′DNA7. For this assay, 100 nM of LbCas12a, 100 nM of crRNA, and 7.4 nM of GFP fragment were used. Error bars represent mean ± SD, where $n = 3$ technical replicates. **f** Time-dependent cis-cleavage of LbCas12a on GFP in the presence of nonspecific ssDNA M13mp18. The reaction mixture was taken out every 5 min and quenched with 6× SDS-containing loading dye. **g** Dissociation constants of crGFP vs. crGFP + 3′DNA7. The Kd was determined by the biolayer interferometry binding kinetic assay with $R2 > 0.9$. Error bars represent ± SD, where $n = 5$ independent dilutions. Source data are available in the Source Data file.

for all the orthologs tested. Overall, LbCas12a showed the highest fluorescence signal, which is consistent with previous studies[13,18]. Through observation of the fluorophore-quencher-based reporter assay and time-dependent gel electrophoresis, we hypothesized that the various extensions of ssDNA on the crRNA induce conformational changes on LbCas12a that result in enhanced endonuclease activity.

Structural analysis of LbCas12a shows that it contains a single RuvC domain, which processes precursor crRNA into mature crRNA, cleaves target dsDNA or ssDNA (referred here as activators), and executes nonspecific cleavage afterward[19,20]. Therefore, we were interested in understanding the effects of these modified crRNAs on cis-cleavage compared to the wild-type crRNA, as well as how cis-cleavage activity correlates to the trans-cleavage activity. Towards this, we carried out an in vitro cis-cleavage assay for various 3′-end and 5′-end modifications. We noticed that the cis-cleavage activity was either similar or marginally improved with most 3′-end modifications, while the 5′-end modifications showed either similar or slightly reduced activity. This phenomenon suggests that the trans-cleavage activity is commensurate with the cis-cleavage activity (Supplementary Figs. 12–13).

**ENHANCE improves the specificity of target detection.** We sought to the characterize specificity of the ENHANCE in discriminating point mutations across dsDNA. By mutating either a single nucleotide or two consecutive nucleotides at each position across the target-binding region, we observed that the crRNA + 3′DNA7 tolerated mutations and produced a stronger fluorescence signal than the wild-type crRNA (crRNA-WT) for both GFP and SARS-CoV-2 targets (Supplementary Figs. 14, 15). As expected, the single-point mutants were more easily tolerated than the double-point mutants by LbCas12a. Nevertheless, it was exciting to note that the fluorescence intensity ratio or the fold-change normalized to the wild-type dsDNA targets was significantly lower for the crRNA + 3′DNA7 compared to wild-type crRNA (Fig. 4 and Supplementary Figs. 14, 15) across both the genes tested. We observed that the 3′DNA7 modifications on crRNAs enhance specificity by up to 8.8-fold across various off-targets when compared to crRNA-WT. Furthermore, based on the statistical analysis, crRNA + 3′DNA7 did not significantly reduce the specificity of detection for the tested targets.

**Divalent cations are crucial for ENHANCE.** Previous studies demonstrated that FnCas12a is a metal-dependent endonuclease, and magnesium ions are required for FnCas12a-mediated self-processing of precursor crRNA[20]. Based on these findings, we hypothesized that different metal ions may significantly affect the trans-cleavage activity of LbCas12a. This led us to test a range of divalent metal cations and discovered that most ions including $Ca^{2+}$, $Co^{2+}$, $Zn^{2+}$, $Cu^{2+}$, and $Mn^{2+}$ significantly inhibited the LbCas12a activity (Supplementary Fig. 16). By further

investigating the $Zn^{2+}$ mediated inhibition of LbCas12a, we found that the inhibition was dose-dependent (Supplementary Fig. 17). Interestingly, $Ni^{2+}$ ions showed an unusual cis-cleavage activity possibly due to its interactions with the His tags on LbCas12a (Supplementary Fig. 16).

Among the tested divalent metal ions, the $Mg^{2+}$ ions showed the highest in vitro cis-cleavage activity, which was consistent with the literature[20]. Therefore, we characterized the effect of $Mg^{2+}$ ions on the trans-cleavage activity of LbCas12a. With increasing the concentration of $Mg^{2+}$ ions, a significant increase in fluorescence signal was observed in an in vitro trans-cleavage assay. By varying the amount of $Mg^{2+}$ in the Cas12a reaction, we identified that the optimal condition of $Mg^{2+}$ was around 13 mM (Fig. 5a, b and Supplementary Figs. 18–21).

**ENHANCE works robustly towards a broad range of targets.** To validate the CRISPR-ENHANCE technology, we first selected a clinically relevant nucleic acid biomarker, Prostate Cancer Antigen 3 (PCA3/DD3), which is one of the most overexpressed genes in prostate cancer tissue and excreted in patients' urine. Consequently, elevated PCA3 level during prostate cancer progression has become a widely targeted biomarker for detection[21–24]. To determine the limit of detection of PCA3 using our ENHANCE technology, we spiked the PCA3 cDNA into synthetic urine and investigated how this clinically relevant environment affects the activity of Cas12a.

Using ENHANCE for detecting the PCA3 cDNA, the limit of detection was determined to be as low as 25 fM in the urine at 13 mM $Mg^{2+}$ concentration compared to ~1 pM at 3 mM $Mg^{2+}$ concentration after 6 h (Fig. 5a–c and Supplementary Figs. 19–21). In contrast, the wild-type crRNA also showed a similar 29 fM limit of detection at 13 mM $Mg^{2+}$ concentration while the limit of detection was ~10 pM at 3 mM $Mg^{2+}$ concentrations after 6 h. Therefore, by combining the crRNA modifications with increased $Mg^{2+}$ ion concentrations, we achieved an approximately 400-fold increase in sensitivity, based on the limit of detection calculations. Nevertheless, this observation also suggests that our modified crRNA + 3′DNA7 significantly improves the limit of detection at low $Mg^{2+}$ but reaches a saturation point that is comparable with the wild-type crRNA at high $Mg^{2+}$ concentration. To understand the importance of divalent ions in the Cas12a trans-cleavage reaction, we carried out a Michaelis-Menten kinetic study with various $Mg^{2+}$ concentrations (Supplementary Fig. 19). We observed that the initial reaction rate of Cas12a in the presence of high $Mg^{2+}$ concentrations increased tremendously compared to that in low $Mg^{2+}$.

However, the two reaction rates eventually reach a similar saturation point (Supplementary Figs. 18–20). This suggests that $Mg^{2+}$ is not only required for the Cas12a reaction but also accelerates the enzyme's trans-cleavage activity. Regardless, $Mg^{2+}$ plays an important role in lowering the limit of detection in synthetic urine containing PCA3. While as low as 25 fM

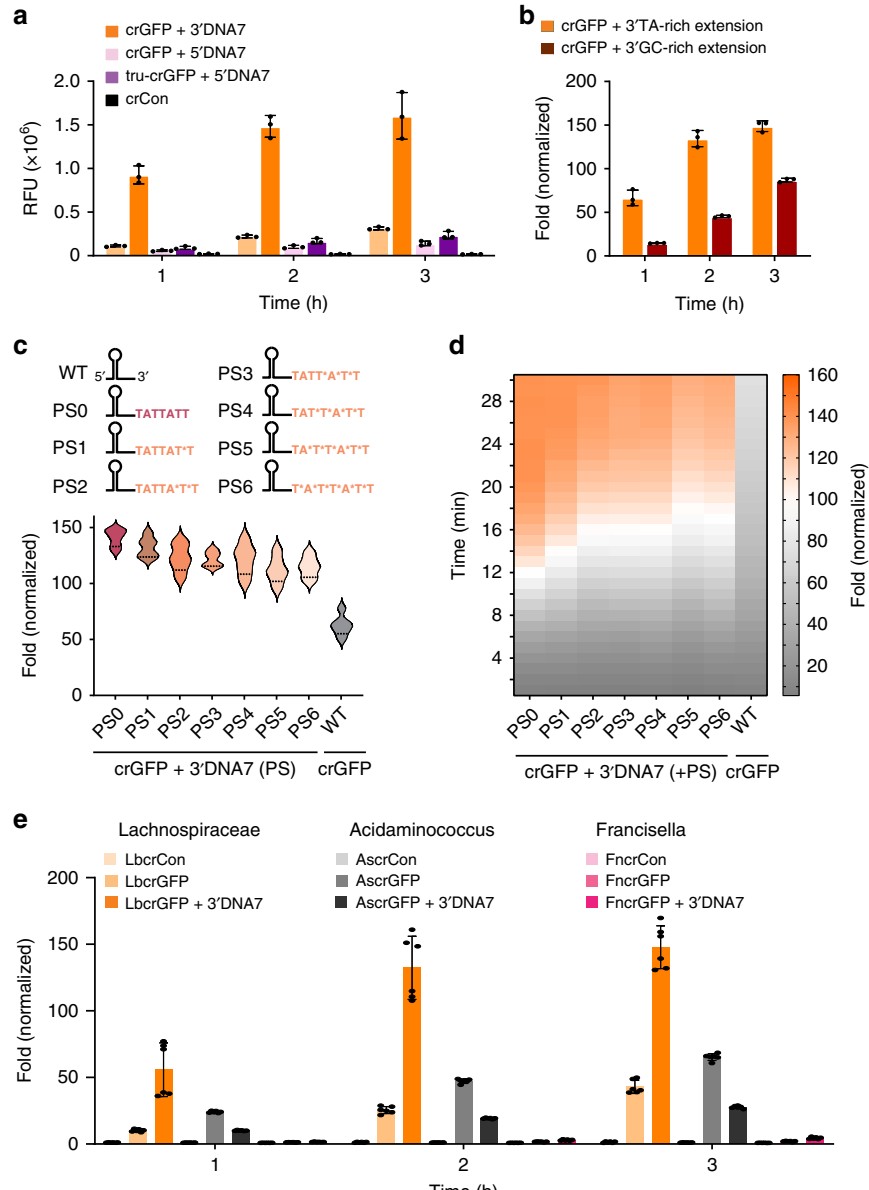

**Fig. 3 Characterization of ENHANCE with various crRNA modifications and different Cas12a systems. a** Comparison of trans-cleavage activity between precursor crRNA (pre-crRNA) and mature crRNA (tru-crRNA, where the first Uracil on the 5'-end of the crRNA is cleaved by LbCas12a in the absence of the activator). **b** Comparison of trans-cleavage activity between AT-rich extensions and GC-rich 7-nt DNA 3'-end extensions on the crRNA + 3'DNA7. For **a**, **b**), error bars denote mean ± SD, where $n = 3$ technical replicates. **c** Trans-cleavage activity of LbCas12a with non-fully phosphorothioate (PS) modified crRNA targeting GFP fragment. Sequence representation of 6 non-fully PS extension on the 3'-end of crGFP ranging from 1 to 6 PS. The asterisk symbol (*) signifies the phosphorothioated nucleotide. The graph below the sequence representation shows fold change of the LbCas12a fluorescence-based reporter assay with the activator normalized to the corresponding samples without the activator at $t = 20$ min. **d** kinetics of the LbCas12a fluorescence-based reporter assay in **c**, **e** Trans-cleavage activity of different variants of Cas12a. The prefix Lb, As, and Fn stand for Lachnospiraceae bacterium, Acidaminococcus, and Francisella novicida, respectively. For **c**, **d**, and **e**, $n = 6$ replicates (three technical replicates examined over two independent experiments), where error bars in **e** represent mean ± SEM. Source data are available in the Source Data file.

(equivalent to 2.5 amol) of PCA3 cDNA can be detected with ENHANCE without any target amplification (Supplementary Fig. 21), the clinical concentration of PCA3 mRNA in the urine can be lower and therefore may require target pre-amplification[25,26]. Therefore, we incorporated and tested a recombinase polymerase amplification (RPA) step to isothermally amplify the PCA3 cDNA. By combining the RPA step as previously reported[1,7], the concentration of PCA3 cDNA in the urine was detectable down to ~10 aM (1 zmol) with a 2.9-fold signal to noise ratio (Fig. 5d).

While crRNA-LbCas12a has been traditionally used to detect unmodified DNA, the field is missing the knowledge on how the common epigenetic marker, DNA methylation, affects its trans-cleavage activity. DNA methylation is also one of the bacterial defense systems that fight against outside invaders. It would be fascinating to understand how LbCas12a collateral cleavage can recognize methylated DNA targets. This curiosity led us to discover that the wild-type crRNA had significantly reduced activity in detecting methylated DNA, containing 5-methyl cytosine, compared to the unmethylated DNA. However, the

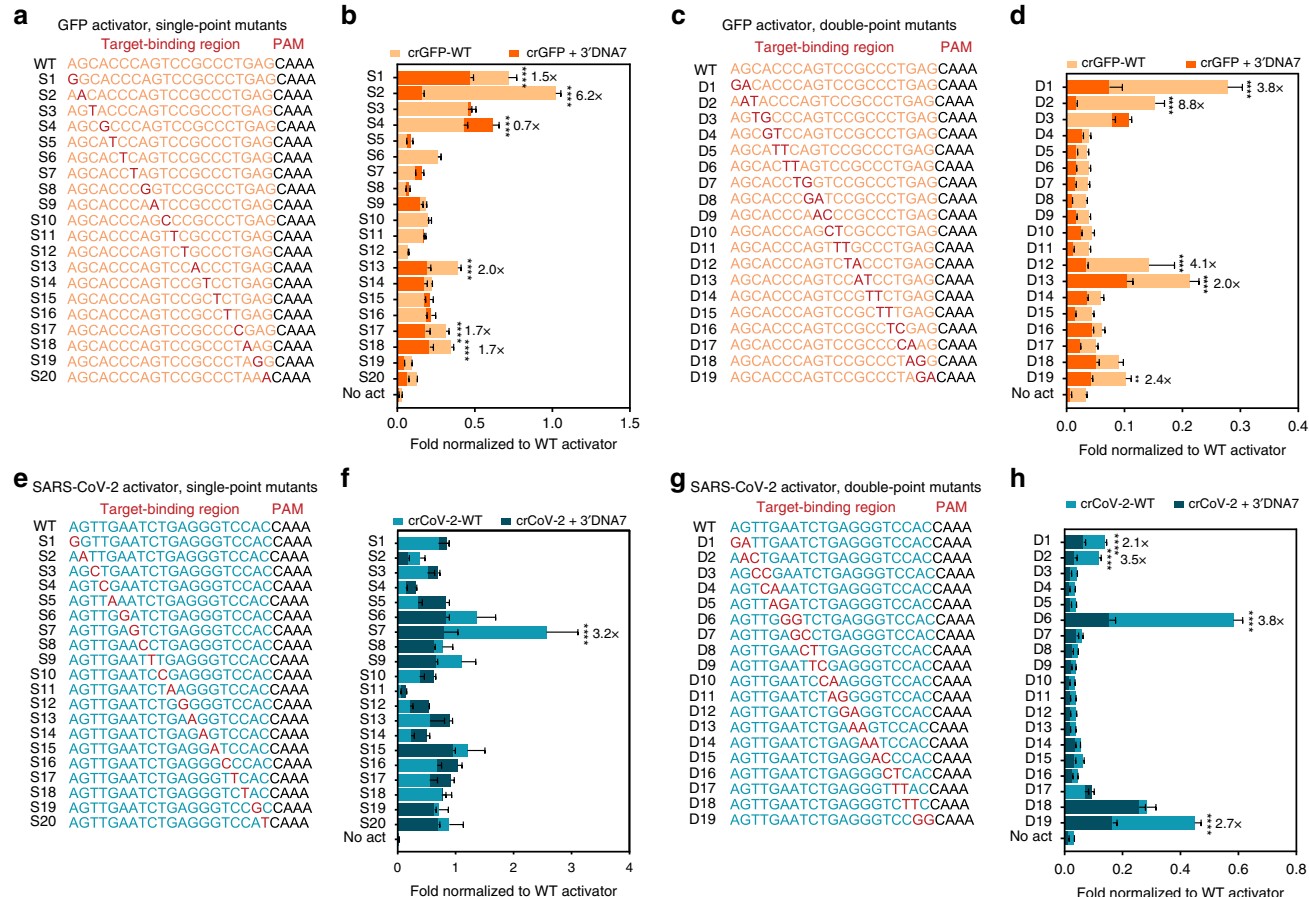

**Fig. 4 Improved specificity of LbCas12a trans-cleavage with ENHANCE. a** Single-point mutations (S1–S20) and **c** double-point mutations (D1–D19) on the target strand of a dsDNA GFP activator. **b** Superimposed bar graphs indicating fold change in fluorescence of the S1–S20 mutant GFP activators normalized to the corresponding wild-type activator **a**. **d** Superimposed bar graphs indicating fold change in fluorescence of the D1–D19 mutant GFP activators normalized to the corresponding wild-type activator in **c**. **e** Single-point mutations (S1–S20) and **g** double-point mutations (D1–D19) on the target strand of a dsDNA SARS-CoV-2 activator. **f** Superimposed bar graphs indicating fold change in fluorescence of the S1–S20 mutant SARS-CoV-2 activators normalized to the corresponding wild-type (WT) activator in **e**. **h** Superimposed bar graphs indicating fold change in fluorescence of the D1–D19 mutant SARS-CoV-2 activators normalized to the corresponding wild-type activator in **g**. All the values **b**, **d**, **f**, and **h** were plotted after 20 min of incubation of various activators with wild-type CRISPR or ENHANCE. For **b**, $n = 6$ (three technical replicates examined over two independent experiments). For **d**, **f**, and **h**, $n = 4$ (two technical replicates examined over two independent experiments). For **b**, **d**, **f**, and **h**, values indicate mean ± SEM and the statistical analysis was performed using two-way ANOVA test with Dunnett's multiple comparison test and only significant ($p < 0.05$) values were marked with an asterisk (*) indicated as follows: *$p < 0.05$, **$p < 0.01$, ***$p < 0.001$, and ****$p < 0.0001$. A fold change in specificity was calculated and reported for only statistically significant mutants by taking the ratio of the normalized data for crRNA-WT to crRNA-3′DNA7. Source data are available in the Source Data file.

ENHANCE showed 5.4-fold and 3.4-fold and higher trans-cleavage activity compared to the wild-type crRNA for targeting the methylated dsDNA and ssDNA, respectively (Fig. 5e and Supplementary Fig. 22a).

Although there are no reports on RNA-guided RNA targeting by LbCas12a, we envisioned that an RNA can potentially be detected as a DNA/RNA heteroduplex. To test this hypothesis, we incorporated a reverse transcription step to convert RNA into cDNA/RNA heteroduplex before detecting the RNA with a trans-cleavage assay. We discovered that the RNA can only be detected if the target strand for crRNA is a DNA but not an RNA in a heteroduplex. Notably, the efficiency of the trans-cleavage activity for the DNA/RNA heteroduplex was found to be significantly lower than the corresponding ssDNA or dsDNA (Fig. 5e, Supplementary Fig. 22b). However, the DNA/RNA heteroduplex achieved an improved enzymatic collateral activity when using the crRNA + 3′DNA7 compared to the wild-type crRNA. We applied the ENHANCE to successfully detect low picomolar concentrations of HIV RNA target encoding Tat gene with our DNA/RNA heteroduplex detection strategy (Fig. 5f). In parallel,

ssDNA and dsDNA targets from HIV were also detected with much higher sensitivity compared to the wild-type crRNA within 15 to 30 min (Fig. 5f, g and Supplementary Fig. 23). We further applied the ENHANCE for detecting HCV ssDNA and HCV dsDNA gene encoding a polyprotein precursor, both of which indicated consistent enhanced collateral activity than the wild-type crRNAs within 24 min (Fig. 5f–h and Supplementary Fig. 24). The limit of detection for HIV and HCV targets was calculated to be 700 fM cDNA and 290 fM ssDNA, respectively.

**ENHANCE detects SARS-CoV-2 genomic RNA with high sensitivity.** In the wake of the recent COVID-19 pandemic, there is an urgent need to rapidly detect the SARS-CoV-2 coronavirus (referred to as CoV-2 here for simplicity). We optimized the ENHANCE to detect CoV-2 dsDNA by designing crRNAs targeting nucleocapsid phosphoprotein encoding N gene (Fig. 3f, i). While no clinical samples were tested, the results indicated the 3′ DNA7-modified crRNA consistently demonstrated higher sensitivity for detecting CoV-2 dsDNA within 30 min as compared to

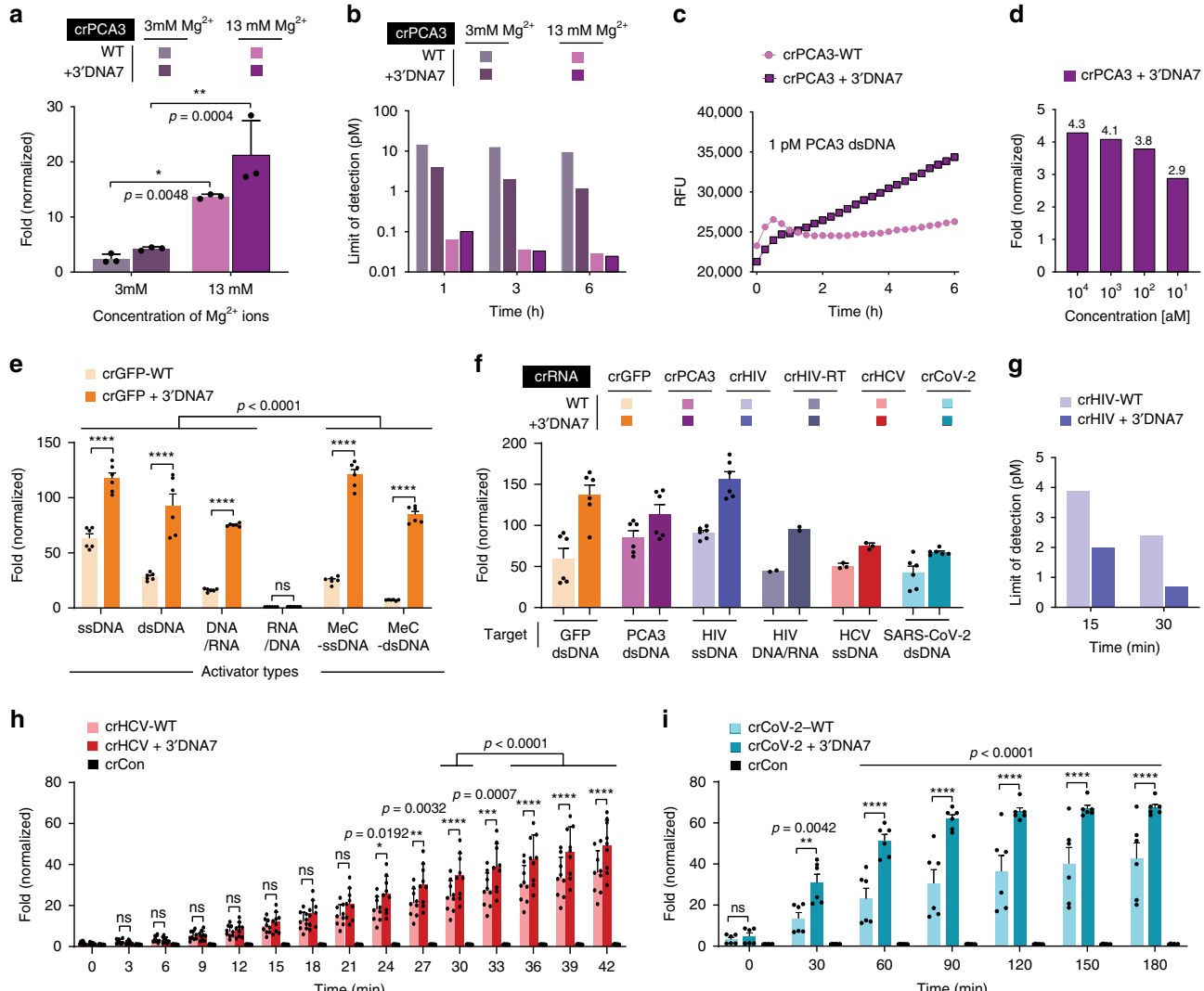

**Fig. 5 Improved detection of various targets using the ENHANCE system. a** Effect of Magnesium ion on the trans-cleavage activity. Error bars represent mean ± SD, where $n = 3$ technical replicates. Statistical analysis was performed using two-way ANOVA test with Sidak's multiple comparison test, where ns = not significant, and the asterisk (*) denotes $p$ values. **b** Limit of detection in femtomolar targeting PCA3 in simulated human urine at optimized Mg2+ concentration. **c** Raw fluorescence data showing detectable signal difference at 1 pM of PCA3 DNA. **d** Fold change in fluorescence signal of the modified crRNA + 3′DNA7 targeting GFP after the recombinase polymerase amplification (RPA) step. **e** Effect of heteroduplex DNA-RNA and methylated activators on the trans-cleavage activity of LbCas12a. **f** trans-cleavage activity of different DNA targets. GFP, PCA3, COVID-19, and HCV are dsDNA activators, and their fluorescence shown were taken after 3 h. The HIV target is ssDNA or cDNA/RNA heteroduplex, and its fluorescence signal shown was taken after 1 h. **g** Limit of detection targeting HIV cDNA fragment after 15 and 30 min. Results in **b** and **g** are based on the limit of detection calculations. **h** Fold change in trans-cleavage activity with LbCas12a in presence of 100 pM (10 fmols) of target HCV ssDNA. Using modified crRNA, the limit of detection of HCV target ssDNA was found to be 290 fM (29 amoles) at 30 min, without target amplification. **i** Fold change in trans-cleavage activity with LbCas12a in presence of 100 pM (10 fmols) of target SARS-CoV-2 cDNA (dsDNA). For **h**, error bars represent mean ± SEM, where $n = 9$ replicates (three technical replicates examined over three independent experiments). For **e**, **f**, and **i**, error bars represent mean ± SEM, where $n = 6$ replicates (three technical replicates examined over two independent experiments). For **e**, **h**, and **i**, statistical analysis was performed using a two-way ANOVA test with Dunnett's multiple comparison test, where ns = not significant with $p > 0.05$, and the asterisks (*, **, ***, ****) denote significant differences with $p$ values listed above. Source data are available in the Source Data file.

the wild-type crCoV-2 (Supplementary Figs. 25–27). By incorporating a commercially available paper-based lateral flow assay with a FITC-ssDNA-Biotin reporter[1,27,28], we could visually detect 1 nM of CoV-2 cDNA at room temperature, 25 °C, within 20 min of incubation using both wild-type and modified crRNAs without any target amplification (Fig. 6 and Supplementary Fig. 28). The enzyme trans-cleavage activity exhibited a consistent trend with the crRNA + 3′DNA7 among five different targets (Fig. 5f). When incorporating a reverse transcription step and a loop-mediated isothermal amplification (RT-LAMP) strategy into

the ENHANCE, both the crCoV-2-WT and the crCoV-2 + 3′ DNA7 demonstrated a limit of detection down to 3–300 copies of RNA (Fig. 6a–c). However, in the case of crCoV-2-WT, the partial cleavage of the reporter resulted in a darker control line on the paper strip. Band-intensity analysis showed that the ENHANCE exhibited an average of 23-fold higher ratio of positive to control line between 1 nM ($3 \times 10^9$ total copies) and 1 pM ($3 \times 10^6$ total copies) of target CoV-2 RNA, while the crCoV-2-WT indicated an average of only 7-fold ratio (Fig. 6a–c and Supplementary Fig. 29).

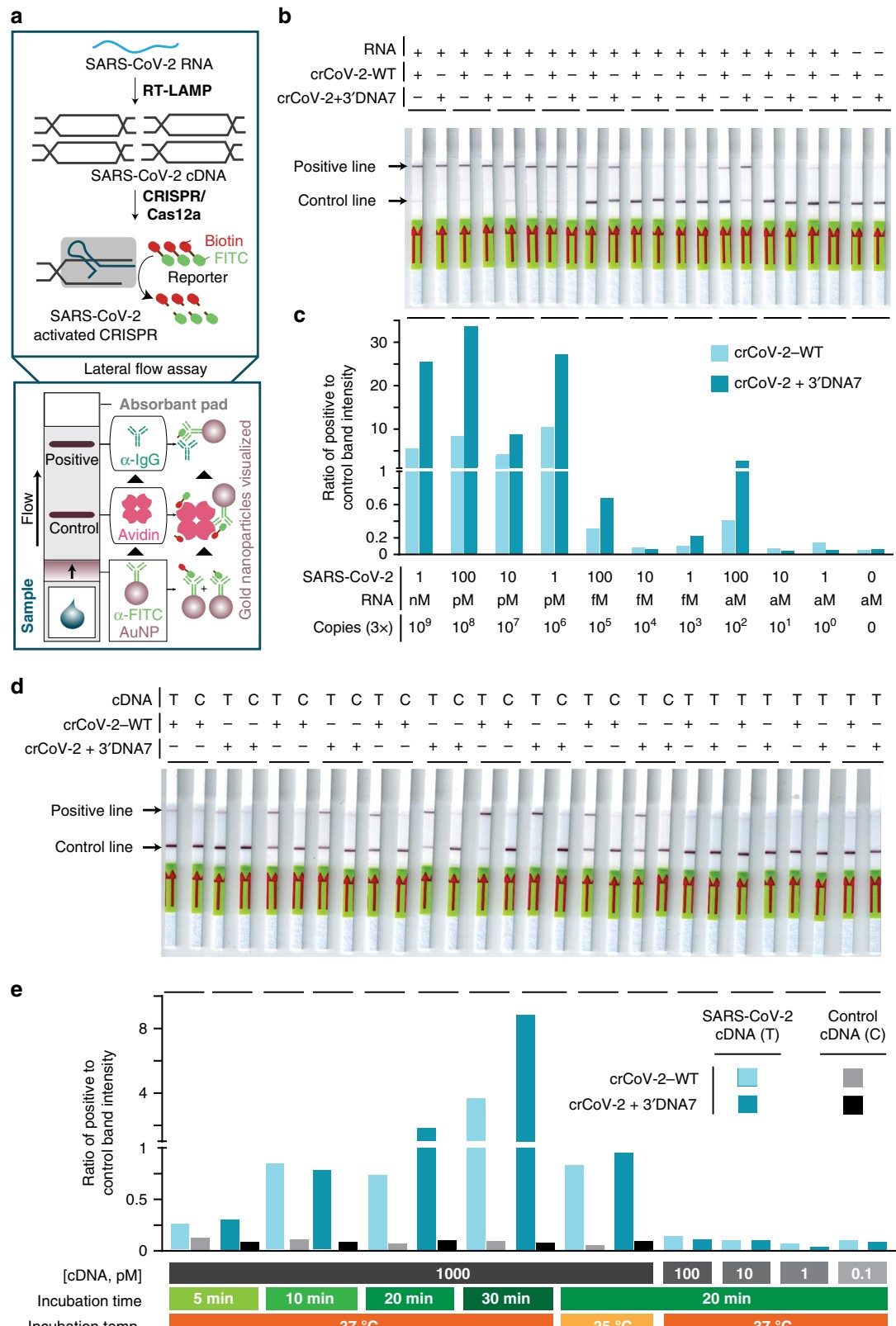

**Fig. 6 Improved detection of COVID-19 in a lateral flow assay using ENHANCE. a** Schematic diagram showing how a lateral flow assay works. Briefly, the dipstick uses gold-labeled FITC-specific antibodies that bind to FITC-biotin reporters and travel through the membrane. Only the cleaved reporters will reside at the positive line. **b** Lateral flow assay detecting SARS-CoV-2 RNA N gene using crCov-2 and crCoV-2 + 3′DNA7 with RT-LAMP, and **c** band-intensity analysis of **b** using ImageJ. **d** Lateral flow assay detecting SARS-CoV-2 cDNA using crCov-2 and crCoV2 + 3′DNA7 without a pre-amplification step, and **e** band-intensity analysis of **d** using ImageJ. Source data are available in the Source Data file.

We observed a much higher fluorescence intensity when using ENHANCE than the unmodified CRISPR in a very short amount of time, within 10 min, for detecting targets. When we applied the system on a lateral flow assay, the positive band was visible only after 30 s, whereas it took over 1 min to show up using the unmodified crRNA. This suggests that we can utilize the engineered ENHANCE system for much rapid detection of nucleic acids, including SARS-CoV-2 (Supplementary Fig. 30).

We investigated the specificity of the ENHANCE by testing crRNAs programmed to target SARS-CoV-2 against coronaviruses, such as MERS-CoV, SARS-CoV, bat-SL-CoVZC45, and HCoV-NL63. Two guide RNAs were employed to target two different regions of the SARS-CoV-2 N-gene (referred to as N1 and N2 regions). The N1 region of SARS-CoV2 was selected to have ≤2 sequence mismatches with SARS-CoV and bat-SL-CoVZC45. This target region was therefore used to recognize if SARS-like coronaviruses strains are detected. The region N2 was selected from Broughton et al.[28] that was specific for SARS-CoV-2 for exclusivity testing (Fig. 7a). We first targeted SARS-CoV-2, MERS-CoV, and bat-SL-CoVZC45 plasmid controls (purchased from IDT) using these two crRNAs. The engineered N1:crCoV2 + 3′DNA7 and N2:crCoV2 + 3′DNA7 showed 3-fold and 7.8-fold higher in fluorescence signal compared to the wild-type N1:crCoV2-WT and N2:crCoV2-WT after 10 min of incubation, respectively. Notably, the engineered N1:crCoV2 + 3′DNA7 exhibited lower in fluorescence signal against MERS-CoV and bat-SL-CoVZC45, demonstrating 74% enhanced specificity towards SARS-CoV-2 (Fig. 7b, c). We next tested the two guides with clinically relevant extracted genomic RNAs of SARS-CoV-2, SARS-CoV Urbani, and HCoV63 (obtained from BEI resources). Both N1:crCoV + 3′DNA7 and N2:

crCoV + 3′DNA7 showed specificity towards SARS-CoV-2 when an RT-LAMP step was applied (Fig. 7d–f). This specificity was because RT-LAMP primers sets were specific for SARS-CoV-2 (Supplementary Figs. 31–33). Collectively, our ENHANCE system successfully retained the sequence matching fidelity when in complex with LbCas12a with enhanced specificity and significantly higher sensitivity compared to the wild-type crCoV2.

## Discussion

We were able to detect very low copies of SARS-CoV-2 in both fluorescence-based and paper-based lateral flow assay platforms. When detecting the samples with low copies, we observed that unmodified CRISPR exhibited a very small sensitivity ratio between the activator positive and the activator negative samples, which led to difficulty in distinguishing if the target dsDNA was present in these samples. However, with our ENHANCE, the activator positive samples displayed a very intense signal compared to activator negative samples, confirming a higher signal to noise ratio. The 7-mer DNA extension to crRNA is universal and spacer-independent, which means that it can be added to any crRNA without affecting the fidelity of the CRISPR/Cas12a system or significantly affecting the cost of synthesis.

In summary, we extended the 3′-end and 5′-end of the crRNA and discovered an amplified trans-cleavage activity and improved specificity of LbCas12a when the 3′-end is extended with DNA or RNA. We applied this modified crRNA-LbCas12a system with the optimal conditions to detect PCA3 in simulated urine with high sensitivity. This ENHANCE technology enabled us to detect the DNA:RNA heteroduplex and methylated DNA with unprecedented sensitivity. We further employed this system to test a range of target

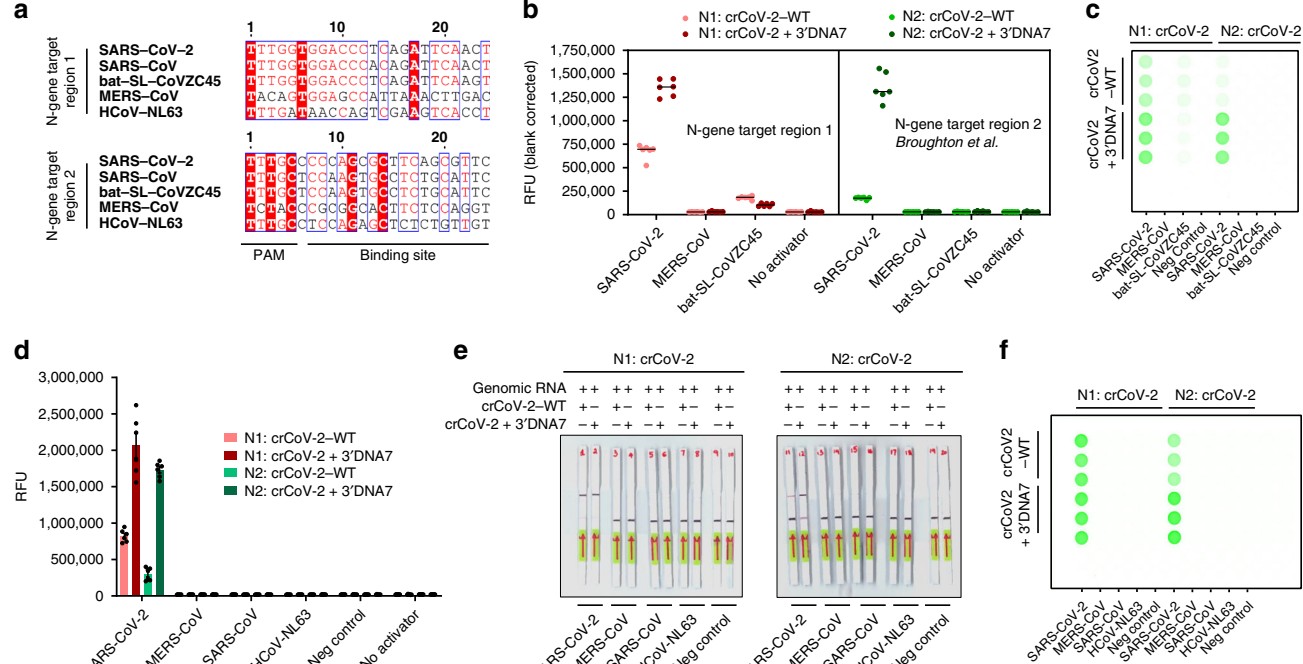

**Fig. 7 Enhanced sensitivity and specificity with ENHANCE for detecting SARS-CoV-2 genomic RNA. a** Sequence alignment of similar pathogens from the same family with SARS-CoV-2 that were tested in this study. Two crRNAs including their engineered version were designed to target two regions of the N gene N1 and N2 (N1: crCoV and N2: crCoV) where N2 was reported in Broughton et al. Sequences were aligned using ClustalOmega[31,32], exported in aln file and graphically enhanced in ESPript 3.0[33]. **b** crRNA specificity towards SARS-CoV-2 and other highly similar pathogens from the same family. The targets were dsDNA amplified from plasmid controls 2019-nCoV_N_Positive Control, MERS-CoV Control, and SARS-CoV Control (IDT). **c** Detection reaction in **b** scanned by Typhoon (Amersham, GE healthcare). **d** crRNA specificity towards genomic RNA of SARS-CoV-2 and other genomic RNAs of highly similar pathogens from the same family. The targets were genomic RNA obtained from BEI Resources. **e** Lateral flow assay of **d**. **f** Detection reaction in **d** scanned by Typhoon (Amersham, GE healthcare). For **b**, and **d**, error bars represent mean ± SEM, where n = 6 replicates (three technical replicates examined over two independent experiments). Source data are available in the Source Data file.

nucleic acids, including ssDNA, dsDNA, and RNA from HIV, HCV, and SARS-CoV-2 without the need for further optimization. Compared to wild-type CRISPR, the ENHANCE demonstrated enhanced detection of SARS-CoV-2 genomic RNA with improved sensitivity and specificity in a fluorescence-based assay and a paper-based lateral flow assay. Although no patient samples were tested in this study, these findings are a crucial step towards enhancing the detection of nucleic acids that can potentially assist in the diagnosis of various diseases including the COVID-19.

## Methods

**DNA activator preparation**. Multiple DNA activators were used in this study. The GFP fragment (942 bp) was produced by amplifying the pEGFP-C1 plasmid using a polymerase chain reaction in the Proflex PCR system (ThermoFisher Scientific). The PCR product was purified using Monarch® Nucleic Acid Purification Kit (New England Biolabs Inc.).

In addition, the 40-nt ds-GFP, ds-PCA3, ds-HCV, ds-HIV, ds-CoV2, and their respective mutants activators were generated by annealing two single-stranded TS and NTS fragments at a 1:1 ratio (Integrated DNA Technologies Inc.) in 1× hybridization buffer (20 nM Tris-Cl, pH 7.8, 100 mM KCl, 5 mM MgCl2). The annealing process was executed in the Proflex PCR system at 90 °C for 2 min followed by gradual cooling to 25 °C at a rate of 0.1 °C/s.

**LbCas12a expression and purification**. The plasmid LbCpf1-2NLS (Addgene #102566, a gift from Jennifer Doudna Lab)[29] was transformed into Nico21(DE3) competent cells (New England Biolabs). Colonies were picked and inoculated in Terrific Broth at 37 °C until OD600 = 0.6. IPTG was then added to the cultures, and they were grown at 18 °C overnight.

Cell pellets were collected from the overnight cultures by centrifugation, resuspended in lysis buffer (2 M NaCl, 20 mM Tris-HCl, 20 mM imidazole, 0.5 mM TCEP, 0.25 mg/ml lysozyme, and 1 mM PMSF, pH = 8), and broken by sonication. The sonicated solution then underwent high-speed centrifugation at 40,000 RCF for 45 min. The collected supernatant was then run through a Ni-NTA Hispur column (Thermofisher) pre-equilibrated with wash buffer A (2 M NaCl, 20 mM Tris-HCl, 20 mM imidazole, 0.5 mM TCEP, pH = 8). The column was then eluted with buffer B (2 M NaCl, 20 mM Tris-HCl, 300 mM imidazole, 0.5 mM TCEP, pH = 8). The eluted fractions were then pooled together and underwent TEV cleavage overnight at 4 °C (TEV protease was purified using the plasmid pRK793, #8827 from Addgene, a gift from David Waugh Lab).

The resulting fraction was equilibrated with buffer C (100 mM NaCl, 20 mM HEPES, 0.5 mM TCEP, pH = 8) at a 1:1 ratio and run through Hitrap Heparin HP 1 ml column (GE Biosciences). The column was washed with buffer C and gradually eluted at a gradient rate with buffer D (100 mM NaCl, 20 mM HEPES, 0.5 mM TCEP, pH = 8). The eluted fraction was concentrated down to 500 μl and passed through the Hiload Superdex 200 pg column (GE Biosciences). The purified LbCas12a was then buffer exchanged with storage buffer (500 mM NaCl, 20 mM Na2CO3, 0.1 mM TCEP, 50% glycerol, pH = 6) and flash-frozen at −80 °C until use.

**Biolayer Interferometry (BLI) binding kinetic assay**. The BLI Ni-NTA biosensors were purchased from Fortebio to perform the binding kinetic study with polyhistidine-tagged LbCas12a. In detail, the experiment was carried out in a 96-well plate and included five steps: baseline, loading, baseline2, association, and dissociation. The biosensors were dipped into the baseline containing 1× kinetic buffer (1× PBS, 0.1% BSA, and 0.01% Tween 20). They were then transferred into each loading well containing 10 μg/ml of LbCas12a. After processing through loading and baseline2, the protein-tagged biosensor was next allowed to dip into the crRNA sample wells at different dilutions (10, 5, 2.5, 1.25, 0.625, 0.3125, 0.15625, and 0 μg/ml) in the association step. The dissociation step occurred when the biosensors were transferred back to baseline2 at a shaking speed of 1000 rpm. All the samples were read by the Octet QKe system (Fortebio). $K_d$ was determined by software Data Analysis 10.0 (Fortebio), and only $K_d$ with $R^2 > 0.9$ were extracted for comparison between crRNA wild-type and modified crRNAs.

**Cis-cleavage assay**. In-vitro digestion reactions were carried out with three different types of the Cas12a family (LbCas12a, AsCas12a, and FnCas12a were purchased from New England Biolabs Inc. or purified in the lab, Integrated DNA Technologies Inc., and abm®, respectively) and a wide array of modified crRNAs (purchased from DNA Technologies Inc.). Cas12a and crRNA were mixed with a 1:1 ratio (100 nM:100 nM) in 1× NEBuffer 2.1 and pre-incubated at 25 °C for 15 min to promote the ribonucleoprotein complex formation. DNA activator (final concentration of 7 nM) was then added to the mixture to produce a total volume of 30 μl and incubated at 37 °C for 30 min[19]. The sample was then analyzed in either 1% agarose gel (for GFP fragments amplified from the pEGFP-C1 plasmid) pre-stained with either SYBER Gold (Invitrogen), GelRed (Biotium Inc.), or premade 15% Novex™ TBE-Urea Gel (Invitrogen).

**M13mp18 nonspecific cleavage assay**. Nonspecific cleavage activity of Cpf1 was activated by incubating Cpf1, crRNA, and DNA activator with a concentration of 100 nM:100 nM: 2 nM in 1× NEBuffer 2.1 buffer at 37 °C for 30 min. M13mp18 was then added to the 30 μl reaction mixture and incubated for an additional 45 min. A fraction of the reaction was taken out every 5 min, quenched in 6× purple gel loading dye (New England Biolabs Inc.), and subsequently analyzed in 1% agarose gel (Fisher Scientific)[1].

**Trans-cleavage reporter assay**. The fluorophore-quencher reporter assay was carried out following a standard clinical detection protocol. The Cas12a-crRNA ribonucleoprotein complex was assembled by mixing 100 nM Cas12a and 100 nM crRNA in 1× NEBuffer 2.1 in the Proflex PCR system (ThermoFisher Scientific) at 25 °C for 15 min (volume 28.5 μl). The activator (1 nM final concentration), FQ reporter (500 nM final concentration), and UltraPure™ DNase/RNase-Free distilled water (Invitrogen) were pre-added to a 96-well plate (Greiner Bio-One) to a volume of 71.5 μl. The reaction was initiated by adding the Cas12a-crRNA mixture to the 96-well plate preloaded with activator and FQ reporter (Integrated DNA Technologies Inc). The plate was quickly transferred to a plate reader (ClarioStar or BioTek), and fluorescence intensity was measured every 3 min for 3 or 12 h (detection limit assay) (FAM FQ: $\lambda_{ex}$: 483/30 nm, $\lambda_{em}$: 530/30 nm; HEX: $\lambda_{ex}$: 430/20 nm, $\lambda_{em}$: 555/30 nm). After 3 or 12 h (detection limit assay), the sample was scanned for images using the Amersham Typhoon (GE Healthcare).

For Michaelis–Menten kinetic study, 30 nm LbCas12a: 30 nM crRNA: 1 nM activator were mixed in NEBuffer 2.1 and incubated at 37 °C for 30 min. The reaction mixture was then transferred to a 96-well plate (Greiner Bio-One) preloaded with different concentrations of FQ reporter (HEX or FAM FQ reporter: 0 M, 0.05 μM, 0.1 μM, 0. 25 μM, 0.5 μM, and 1 μM) and UltraPure™ DNase/RNase-Free distilled water (Invitrogen)[1].

To find the limit of detection (LoD), the fluorophore-quencher reporter assay was carried out with various concentrations of activator. The LoD calculations were based on the following formula[30]:

$$LoD = \frac{3.3 \times \text{Std of RFU in the absence of activator}}{\text{Slope of RFU vs. Activator concentration}}$$

**Effects of metal ions on Cas12a cleavage study**. The metal ions ($Mg^{2+}$, $Zn^{2+}$, $Mn^{2+}$, $Cu^{2+}$, $Co^{2+}$, $Ca^{2+}$) were prepared by diluting chloride salt in different concentrations. For cis-cleavage, the Cas12a-crRNA-metal iron duplex was mixed with 100 nM: 100 nM: varying nM ratio in 1× annealing buffer (100 mM Tris-HCl, pH 7.9 at 25 °C, 500 mM NaCl, 1 mg/ml BSA) and pre-incubated at 25 °C for 15 min. DNA activator (GFP or PCA3 fragments) was then added to the mixture to a total volume of 30 μl and incubated at 37 °C for 30 min.

**Paper strip test**. To minimize the testing time, the following reagent was assembled in a one-pot reaction:

a. 10× NEBuffer: 5 μl (1× final concentration)
b. 3 μM LbCas12a: 1 μl (60 nM final concentration)
c. 3 μM crRNA: 2 μl (120 nM final concentration)
d. 5 μM FAM-biotin reporter: 1.5 μl (150 nM final concentration)
e. Various concentration of activator: 2 μl
f. Nuclease-free water: 38.5 μl (total reaction volume = 50 μl)

The reaction mixture was incubated at either 37 °C or 25 °C for different durations (5, 10, and 20 min). A Milenia HybriDect (TwistDx) strip was dipped in each reaction and allowed for rapid visualization.

For experiments involving a recombinase polymerase amplification (RPA) step, the reaction mix was prepared in the following order:

a. Forward primer (10 μM): 2.4 μl
b. Reverse primer (10 μM): 2.4 μl
c. Primer free rehydration buffer: 29.5 μl
d. Template and nuclease-free water: 13.2 μl
e. (CH3COO)2Mg (280 mM): 2.5 μl (total volume = 50 μl)

The RPA reaction was incubated at 39 °C for 20–30 min before the LbCas12a reaction.

For experiments involving an RT-LAMP preamplification step of target RNA, the mixture was prepared in the following order (except for the RNA and primer mix samples (IDT Technologies), everything was purchased from New England Biolabs):

a. 10× isothermal amplification buffer: 2.5 μl
b. 100 mM MgSO4: 1 μl
c. 10 mM dNTP: 3.5 μl
d. 20× primer mix (4 μM F3, 4 μM B3, 32 μM FIP, 32 μM BIP, 16 μM LF and 16 μM BF): 1.25 μl
e. Bst 2.0 polymerase: 1 μl
f. Warmstart RTx: 0.5 μl
g. RNase inhibitor, murine (40,000 U/ml): 0.625 μl
h. Nuclease-free water: 9.625 μl
i. RNA sample: 5 μl (total volume = 25 μl)

The RT-LAMP reaction was incubated at 63 °C for 20–30 min prior to LbCas12a reaction.

**Reporting summary**. Further information on research design is available in the Nature Research Reporting Summary linked to this article.

## Data availability

All the data supporting the findings of this study are available within the Article and Supplementary Files or can be obtained from the corresponding author, P.K.J., upon reasonable request. Source data are available in the Source Data file. Source data are provided with this paper.

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

## Acknowledgements

We are grateful to the members in the Jain lab for their helpful discussions and the University of Florida (UF) Health Cancer Center for their support. We are particularly thankful to Eric Beck for editing the manuscript and Ling Jin, Santosh Rananaware, and Marco Downing for helping with the experiments and/or data analysis. We also thank the Monoclonal Antibody core facility staff, especially Dr. Angle Sampson and Shadi Bootorabi, at the UF Interdisciplinary Center for Biotechnology Research (ICBR) for coordinating the biolayer interferometry experiments. This research was supported by the internal funding from the UF and the UF Herbert Wertheim College of Engineering. The following reagents were obtained through BEI Resources, NIAID, NIH: Genomic RNA from the Middle East Respiratory Syndrome Coronavirus (MERS-CoV), EMC/2012, NR-45843; Quantitative PCR (qPCR) Control RNA from Inactivated SARS Coronavirus, Urbani, NR-52346; Genomic RNA from Human Coronavirus (HCoV), NL63, NR-44105. The following reagent was deposited by the Centers for Disease Control and Prevention and obtained through BEI Resources, NIAID, NIH: Genomic RNA from SARS-Related Coronavirus 2, Isolate USA-WA1/ 2020, NR-52285.

## Author contributions

P.K.J. initiated the study; L.T.N. and P.K.J. designed research; L.T.N. and B.M.S. performed research; L.N.T., B.M.S., and P.K.J. analyzed the data; L.N. and P.K.J. wrote the manuscript that was edited and approved by all authors.

## Competing interests

P.K.J., L.T.N., and B.M.S. are listed as inventors on the following patent applications related to the content of this work: (1) Highly sensitive nucleic acid detection using modified CRISPR/Cas. Applicant: University of Florida. Inventors: P.K.J. and L.T.N. Status: Patent pending. US Patent App. 62/932,823; US Patent App. 62/988,679, and US Patent App. 63/001,056. Increasing Cas-mediated trans-cleavage activity and enhancing the detection of nucleic acid targets by extending crRNAs. (2) Engineered CRISPR/Cas systems for detecting DNA/RNA heteroduplexes and methylated DNA. Applicant: University of Florida. Inventors: P.K.J., L.T.N., and B.M.S. Status: Patent pending. US Patent App. 62/ 952,762. Application of extended crRNAs (CRISPR-ENHANCE) for detecting an RNA via DNA/RNA heteroduplex and methylated DNA. (3) Trans-cleavage activity of Cas12a with engineered crRNA enables amplified nucleic acid detection. Applicant: University of Florida. Inventors: P.K.J. and L.T.N. Status: Patent pending. US Patent App. 63/010,382. Enhancement of trans-cleavage activity by nucleic acid extensions on the crRNA. Application of CRISPR-ENHANCE for detecting SARS-CoV-2 genetic material.
