## [Peer Review File · Nature Communications]

Reviewers' Comments:

Reviewer #1:

Remarks to the Author:

In this manuscript entitled " Enhancement of trans-cleavage activity of Cas12a with engineered crRNA enables amplified nucleic acid detection" by Nguyen et al., the authors find that 3' end extension of the Cas12a crRNA enhances trans-cleavage activity and it can be applied to nucleic acid detection. This work shows some unique findings that are of interest of the field and demonstrates timely application of the technology for Covid-19 detection.

Authors conducted extensive experiments and discovered a number of interesting facts regarding LbCas12a.

1. My major concern is how significantly 3' end extension affects its accuracy. Authors have demonstrated clearly that sensitivity increases with 3' end extension. However, accuracy is still not fully demonstrated. 3' end is a part that directly scrutinize sequence match. There is a chance that extension with 7 nt DNA affects target sequence recognition and increase off-target cutting. Figure 2k and Supplemental Figure 13 (by the way, others are supplementary figure and this is supplemental) answer a little bit of sensitivity issue. I do not understand how the authors drew the conclusion mentioned in main text. To me, it looks like crGFP+3'DNA7 seems to cleave dsDNA with mutation better than crGFP. Additional explanation or experiments can help.

2. Regarding the detection capability, it would be great to see a bit more discussion. What does increase of this much intensity mean? How much does it simplify the process? How the detection limits are comparable to other systems? Covid-19 diagnosis is a huge issue. There are other published works like Broughton et al. Nature Biotech 2020. Would you be able to compare with other systems?

Minor comments

3. Figure 3c. crPCA3 WT graph is interesting. It would be great if you can comment a bit more about it

4. Figure 3 k and 3m are a bit hard to read. Maybe figure can be cropped and have it bigger will be helpful to readers

5. One of supplementary figures uses Cpf1 instead of Cas12a. It will be great to mention in the manuscript to clean up the nomenclature.

I want to emphasize that authors did great job.

Reviewer #2:

Remarks to the Author:

In this manuscript, the authors present data demonstrating that 3' modification of Cas12a crRNA enhances trans-cleavage activity. Specifically, they show that a 3' "DNA 7" extension significantly enhances activity as compared to unmodified crRNA. The authors then showed that 3' end processing of crRNA is activator dependent and provide interesting mechanistic evidence based of published structural data. Moving towards developing a highly sensitive CRISPR based detection assay, the authors looked at the contribution of divalent metal cations to enzymatic activity. Consistent with the literature, they showed that LbCas12a to be Mg²⁺ sensitive and optimized its concentration in their trans-cleavage activity assay. They thus developed CRISPR-ENHANCE and used detection of Prostate Cancer Antigen 3 as a proof-of-concept experiment. Interestingly, the authors show that ENHANCE can target 5-methyl cytosine DNA with 3 to 5-fold higher sensitivity compared to wild-type crRNA. In light of current events, the authors merged ENHANCE with a commercially available paper-based lateral flow assay to visualize detection of SARS-CoV-2 cDNA in 20 minutes without target amplification. When paired with RT-LAMP, their ENHANCE system displayed a 23-fold higher sensitivity.

Points to consider:

- Figure 2k: This could be mentioned in the main text and presented as a supplemental figure. This analysis hints at representing an overall picture of target strand engagement. The authors may find that single-point mutation-activity profiles can vary greatly depending on the target.
- It would be interesting to present data using non-fully PS modified DNA. For example, 3'PSDNA7 using 1 to 7 PS substitutions
- Along the same lines, 2'-DNA and 3'-RNA modifications studies would be a nice addition to this work.
- Moving forward as a detection tool, assay selectivity and discrimination are critical. Data demonstrating that ENHANCE can discriminate between SARS-CoV, MERS-CoV and SARS-Cov2 for instance, would really strengthen the manuscript.

Enhancement of trans-cleavage activity of Cas12a with engineered crRNA enables amplified nucleic acid detection

We truly appreciate the reviewers for their valuable feedback regarding our study of the engineered CRISPR-Cas12a system. We have addressed all the comments from the reviewers below (changes to the manuscripts are highlighted in yellow):

Reviewer #1 (R1):

In this manuscript entitled " Enhancement of trans-cleavage activity of Cas12a with engineered crRNA enables amplified nucleic acid detection" by Nguyen et al., the authors find that 3' end extension of the Cas12a crRNA enhances trans-cleavage activity and it can be applied to nucleic acid detection. This work shows some unique findings that are of interest of the field and demonstrates timely application of the technology for Covid-19 detection.

Authors conducted extensive experiments and discovered a number of interesting facts regarding LbCas12a.

Response to R1- We thank the reviewer for general insight of the manuscript.

R1.1 My major concern is how significantly 3' end extension affects its accuracy. Authors have demonstrated clearly that sensitivity increases with 3'end extension. However, accuracy is still not fully demonstrated. 3' end is a part that directly scrutinize sequence match. There is a chance that extension with 7 nt DNA affects target sequence recognition and increase off-target cutting. Figure 2k and Supplemental Figure 13 (by the way, others are supplementary figure and this is supplemental) answer a little bit of sensitivity issue. I do not understand how the authors drew the conclusion mentioned in main text. To me, it looks like crGFP+3'DNA7 seems to cleave dsDNA with mutation better than crGFP. Additional explanation or experiments can help.

Response to R1.1- We appreciate reviewer comments, and we believe that this is a valid concern. We have thoroughly tested specificity and added two main figures (Fig. 4 and Fig. 7) and two SI figures (Fig. 14 and 15) as suggested.

We agree with reviewer #1 that indeed, the crGFP+3'DNA7 seems to cleave dsDNA with mutation better than crGFP as illustrated in Supplementary Figs. 14 and 15. However, it is more appropriate to compare the ratio of these fluorescence data to the corresponding wild-type target dsDNA for each crRNA to find if the mutants are further changing the specificity with respect to the wild-type target dsDNA. In fact, when normalizing the raw fluorescence signal of all the mutated activators to the wild-type activator, we observed that the crGFP+3'DNA7 slightly enhanced the specificity by giving off a significantly lower (up to 8.8-fold) noise-to-signal ratio (the noise referred to different mutated activators and the signal referred to wild-type activator) compared to the crGFP (Fig. 4 in the main text).

Figure 4: Improved specificity of LbCas12a trans-cleavage with CRISPR-ENHANCE. (a) Single-point mutations (S1-S20) on the target strand of a dsDNA GFP activator. (b) The heat map displays relative fluorescence intensity normalized to wild-type (WT) activator after 3 hours for a pilot study (n=1). (c) Double-point mutations (D1-D19) on the same target dsDNA GFP activator in (a). (d) Superimposed bar graphs indicating fold change in fluorescence of the mutant activators normalized to the corresponding wild-type activator in (c). (e) Single-point mutations (S1-S20) on the target strand of a dsDNA SARS-CoV-2 activator. (f) Superimposed bar graphs indicating fold change in fluorescence of the mutant activators normalized to the corresponding wild-type activator in (e). (g) Double-point mutations (D1-D19) on the same target dsDNA SARS-CoV-2 activator in (e). (h) Superimposed bar graphs indicating fold change in fluorescence of the mutant activators normalized to the corresponding wild-type activator in (g). Error bars in (d), (f), and (h) represent \pm SEM, where n = 6 from 2 separate experiments; two-way ANOVA test two-way ANOVA (n=3, N=2, P > 0.05 (no asterisks), **P < 0.01, ***P < 0.0001). A fold change in specificity was calculated and reported for only statistically significant mutants by taking the ratio of the normalized data for crRNA-WT to crRNA-3'DNA7. The experiments were repeated at least twice with n = 3 per experiment.

To further explore the LbCas12a tolerance towards mutated versions of dsDNA activators when in complex with the modified crRNA+3'DNA7, we conducted 3 additional experiments with n=3 (replicates) and N=2 (repeats) on 19 double-point mutations of the GFP target, 20 single-point and 19 double-point mutations of our new target, SARS-CoV-2. We have added the data to the main text and the SI section (Fig. 4 and Supplementary Figs.14-15) and provided the sequence information of newly tested mutants (116 sequences including sense and antisense strands) in the SI section. From these experiments, we observed that even with single- or double-point mutations, the extended crRNA+3'DNA7 showed less noise-to-signal ratio compared to the unmodified crRNA while exhibiting significantly higher fluorescence signal with the non-mutated activators, which is an indication of higher specificity of our engineered crRNA. While we acknowledged that the accuracy of the crRNA+3'DNA7 was slightly better, the

observations from these experiments led us to conclude that our engineered crRNA did not lose the fidelity of CRISPR/Cas12a binding to its target ssDNA/dsDNA. The main text has been changed to reflect additional experiments as followed (see highlighted text below):

Supplementary Figure 14. Effect of single-point and double-point mutations on the target strand of the double-stranded GFP fragment demonstrated as raw fluorescence signals of Figure 4a-d. (a) Sequence representation of dsDNA GFP WT and single-point mutants. The heat map of (b) the raw fluorescence intensity and (c) the normalized fluorescence intensity with respect to the wild-type activator ($n=1$). The fluorescence data in (b) and (c) were recorded using ClarioStar plate reader after 3 hours of incubation in a pilot experiment. A portion of this data in (c) was plotted in a heat map in the Figure 4b. (d) Sequence representation of dsDNA GFP WT and double-point mutants. (e) Raw fluorescence intensity and (f) the normalized fluorescence intensity with respect to the wild-type activator for 19 double-point mutation of target GFP fragments. The fluorescence data in (e) and (f) were recorded using BioTek Synergy 2 plate reader after 20 minutes, and the error bars represent \pm SEM, where $n = 2$. The same data in (f) was plotted as a bar graph with error bars the Figure 4d. The experiments were repeated at least twice with $n = 2$ per experiment.

Supplementary Figure 15. Effect of single-point and double-point mutations on the target strand of the double-stranded DNA SARS-CoV-2 demonstrated as raw fluorescence signals of Figure 4e-h. **(a)** Sequence representation of dsDNA SARS-CoV-2 WT and single-point mutants. **(b)** Raw fluorescence intensity and **(c)** the normalized fluorescence intensity with respect to the wild-type activator, for 20 single-point mutants of the target SARS-CoV-2. **(d)** Sequence representation of dsDNA SARS-CoV-2 WT and double-point mutants. **(e)** Raw fluorescence intensity and **(f)** the normalized fluorescence intensity with respect to the wild-type activator, after 20 minutes for 19 double-point mutants of the target. All fluorescence data were recorded using BioTek Synergy 2 plate reader, and the error bars represent \pm SEM, where $n = 2$. The same data in **(c)** and **(f)** were plotted as bar graph with error bars in the Figure 4f and Figure 4g, respectively. The experiments were repeated at least twice with $n = 2$ per experiment.

Next, we sought to characterize specificity of these extended crRNAs in discriminating point mutations across dsDNA. By mutating either a single nucleotide or two consecutive nucleotides at each position across the target-binding region, we observed that the crRNA+3'DNA7 tolerated mutations and produced a stronger fluorescence signal than the wild-type crRNA for both GFP and SARS-CoV-2 targets (Supplementary Figs. 14, 15). As expected single point mutants were more easily tolerated than double mutants.

Nevertheless, it was exciting to note that the fluorescence intensity ratio or the fold-change normalized to the wild-type dsDNA targets was significantly lower for the crRNA+3'DNA7 compared to wild-type crRNA (Fig. 4 and Supplementary Figs.14,15) across both the genes tested. We observed that the 3'DNA7 modifications on crRNAs enhance specificity by up to 8.8-fold across various off-targets when compared to crRNA-WT. Furthermore, based on the statistical analysis, crRNA+3'DNA7 did not significantly reduce the specificity of detection for any mutant tested.

Figure 7. Specificity of CRISPR-ENHANCE for detecting SARS-CoV-2 genomic RNA. (a) Sequence alignment of similar pathogens from the same family with SARS-CoV-2 that were tested in this study. Two crRNAs including their engineered version were designed to target two regions of the N gene N1 and N2 (N1: crCoV and N2: crCoV) where N2 was reported in *Broughton et al.* Sequences were aligned using ClustalOmega^{29,30}, exported in aln file and graphical enhanced in ESPrnt 3.0³¹. (b) crRNA specificity towards SARS-CoV-2 and other highly similar pathogens from the same family. The targets were dsDNA amplified from plasmid controls 2019-nCoV_N_Positive Control, MERS-CoV Control, and SARS-CoV Control (IDT). (c) Detection reaction in (b) scanned by Typhoon (Amersham, GE healthcare). (d) crRNA specificity towards genomic RNA of SARS-CoV-2 and other genomic RNAs of highly similar pathogens from the same family. The targets were genomic RNA obtained from BEI Resources. (e) Lateral flow assay of (d). (f) Detection reaction in (d) scanned by Typhoon (Amersham, GE healthcare). Error bars represent \pm SEM, where $n = 6$ replicates. The experiments were repeated at least twice with $n = 3$ per experiment.

In addition, we have also carried out exclusivity experiments that used our CRISPR-ENHANCE to recognize SARS-CoV-2 against other highly similar coronaviruses in the same family such as MERS-CoV, SARS-CoV Urbani, bat-SL-CoVZC45, and HCoV-NL63 (plasmid controls and extracted genomic RNA targets). All the data regarding target specificity against similar pathogens have been presented in the Figure 7 in the main text of our revised manuscript. Furthermore, we have also updated the abstract and conclusions to reflect our new findings.

R1.2. Regarding the detection capability, it would be great to see a bit more discussion. What does increase of this much intensity mean? How much does it simplify the process? How the detection limits are comparable to other systems? Covid-19 diagnosis is a huge issue. There are other published works like Broughton et al. Nature Biotech 2020. Would you be able to compare with other systems?

Response to R1.2- These are great questions regarding the applications of our engineered crRNA towards diagnostics. Throughout the manuscript, we included the wild-type CRISPR system as our control. This system has been developed for detecting nucleic acids by Chen et al., Science 2018 and has been utilized as diagnostics for COVID-19 detection by many including *Broughton et al.*, Nature Biotech 2020. In this manuscript, we focused on the development of CRISPR-ENHANCE technology. Regarding comparing with other systems, we have tested the crRNA targeting SARS-CoV-2 used in *Broughton et al.*, Nature Biotech (referred to as N2:crCoV2-WT in Fig. 7). As expected, our ENHANCE system showed significantly higher sensitivity compared to their wild-type crRNA.

Figure 7. Specificity of CRISPR-ENHANCE for detecting SARS-CoV-2 genomic RNA. (a) Sequence alignment of similar pathogens from the same family with SARS-CoV-2 that were tested in this study. Two crRNAs including their engineered version were designed to target two regions of the N gene N1 and N2 (N1: crCoV and N2: crCoV) where N2 was reported in *Broughton et al.* Sequences were aligned using ClustalOmega^{29,30}, exported in aln file and graphical enhanced in ESPrnt 3.0³¹. (b) crRNA specificity towards SARS-CoV-2 and other highly similar pathogens from the same family. The targets were dsDNA amplified from plasmid controls 2019-nCoV_N_Positive Control, MERS-CoV Control, and SARS-CoV Control (IDT). (c) Detection reaction in (b) scanned by Typhoon (Amersham, GE healthcare). (d) crRNA specificity towards genomic RNA of SARS-CoV-2 and other genomic RNAs of highly similar pathogens from the same family. The targets were genomic RNA obtained from BEI Resources. (e) Lateral flow assay of (d). (f) Detection reaction in (d) scanned by Typhoon (Amersham, GE healthcare). Error bars represent \pm SEM, where $n = 6$ replicates. The experiments were repeated at least twice with $n = 3$ per experiment.

Currently, we are in the process of validating the system with COVID-19 clinical samples, and the results seem promising for a future publication. We have added more discussion about detection capability in the manuscript as you recommended (see highlighted text below).

We observed a much higher fluorescence intensity when using CRISPR-ENHANCE than the unmodified CRISPR in a very short amount of time, within 10 minutes, for detecting targets. When we applied the system on a lateral flow assay, the positive band is visible only after 30 seconds whereas it takes over 1 minute to show up when using the unmodified crRNA. This suggests that we can utilize the engineered CRISPR-ENHANCE system for a much rapid detection of nucleic acids including SARS-CoV-2.

We were able to detect very low copies of SARS-CoV-2 in both fluorescence-based and paper-based lateral flow assay platforms. When detecting the samples with low copies, we observed that unmodified CRISPR exhibited a very small sensitivity ratio between the activator positive and the activator negative samples which led to difficulty in distinguishing if the target dsDNA was present in these samples. However, with our CRISPR-ENHANCE, the activator positive samples displayed a very intense signal compared to activator negative samples, confirming a higher signal to noise ratio. The 7-mer DNA extension to crRNA is universal and spacer-independent, which means that it can be added to any crRNA without affecting the fidelity of the CRISPR/Cas12a system or significantly affecting the cost of synthesis.

Minor comments

R1.3. Figure 3c. crPCA3 WT graph is interesting. It would be great if you can comment a bit more about it.

Response to R1.3- Thanks for pointing it out. Fig. 3c (now Fig. 5c) is a representative graph of raw fluorescence intensity of detecting 1 pM concentration of dsDNA using crPCA3 WT and crPCA3+3'DNA7. With crPCA3 WT, the fluorescence intensity increases initially but due to low activator concentration of 1 pM, the CRISPR trans-cleavage activity slows down after 30 minutes. We measured fluorescence using a Biotek Synergy 2 plate reader every 3 minutes for 6 hours (reported only every 15 minutes for data representation). We hypothesize that each measurement contributes to photobleaching of fluorescent FAM reporters. We observed decreased fluorescence after 30 minutes when there is a lot of fluorescent FAM reporters. We notice that effect is equipment dependent and is more prominent in our plate reader compared to another model we tested (Clariostar). Therefore, in all our figures (except Fig. 3c, now Fig. 5c), we calculated and reported fold change normalized to a negative control without an activator.

R1.4. Figure 3 k and 3m are a bit hard to read. Maybe figure can be cropped and have it bigger will be helpful to readers.

Response to R1.4- We agreed with the reviewer. We have split the Figure 3 into two figures (Fig. 5 and Fig. 6) to make the lateral flow data legible. We have modified the figure captions and descriptions accordingly. Similarly, we have split Figure 2 into two figures (Fig. 2 and Fig. 3) and increased the font size of all the figures to make them legible.

R1.5. One of supplementary figures uses Cpf1 instead of Cas12a. It will be great to mention in the manuscript to clean up the nomenclature.

Response to R1.5- Thank you. We have converted all the “Cpf1” into “Cas12a” to make the nomenclature consistent across the Supplementary Figures. We mentioned the Cas12a in the main text as followed:

“Class 2 CRISPR/Cas (Clustered Regularly Interspaced Short Palindromic Repeats/CRISPR-associated proteins) systems, such as Cas12a (previously referred as Cpf1, subtype V-A) and Cas13a (previously referred as C2c2, subtype VI), are capable of nonspecific cleavage of ssDNA (single-stranded DNA) and RNA, respectively, in addition to successful gene editing”

I want to emphasize that authors did great job.

Response to R1- Thank you so much for your feedback.

Reviewer #2 (R2):

In this manuscript, the authors present data demonstrating that 3' modification of Cas12a crRNA enhances trans-cleavage activity. Specifically, they show that a 3' "DNA 7" extension significantly enhances activity as compared to unmodified crRNA. The authors then showed that 3'end processing of crRNA is activator dependent and provide interesting mechanistic evidence based of published structural data. Moving towards developing a highly sensitive CRISPR based detection assay, the authors looked at the contribution of divalent metal cations to enzymatic activity. Consistent with the literature, they showed that LbCas12a to be Mg²⁺ sensitive and optimized its concentration in their trans-cleavage activity assay. They thus developed CRISPR-ENHANCE and used detection of Prostate Cancer Antigen 3 as a proof-of-concept experiment. Interestingly, the authors show that ENHANCE can target 5-methyl cytosine DNA with 3 to 5-fold higher sensitivity compared to wild-type crRNA.

In light of current events, the authors merged ENHANCE with a commercially available paper-based lateral flow assay to visualize detection of SARS-CoV-2 cDNA in 20 minutes without target amplification. When paired with RT-LAMP, their ENHANCE system displayed a 23-fold higher sensitivity.

Response to R2- We thank the reviewer for an excellent summary.

Points to consider:

R2.1 Figure 2k: This could be mentioned in the main text and presented as a supplemental figure. This analysis hints at representing an overall picture of target strand engagement. The authors may find that single-point mutation-activity profiles can vary greatly depending on the target.

Response to R2.1- We thank the reviewer for the comment. Figure 2k (now Fig. 4a and Supplementary Fig. 14) was also addressed by reviewer #1 in regard to the specificity of the crRNA ENHANCE version that the 3'-end extension could play a role in its sequence matching to the target ssDNA/dsDNA. crRNA ENHANCE specificity was one of the most important aspects we heavily investigated in our study. We agree that with the reviewer that single-point mutation activity profiles are target dependent. Therefore, we have conducted 3 additional experiments with n=2 (replicates) and N=2 (repeats) on 19 double-point mutations of the GFP target, 20 single-point and 19 double-point mutations of our new target, SARS-CoV-2. The fluorescence-based data collectively suggested that our crRNA ENHANCE version slightly improved the specificity of on-target binding and at the same time, greatly enhanced the trans-cleavage activity of LbCas12a. Additionally, based on later comments, we have tested the specificity of our engineered crRNAs programmed to target SARS-CoV-2 against various SARS-like coronaviruses. We believe that the specificity data are now more robust than before because of thorough testing of multiple crRNAs and mutants with replicates and repeated experiments. Therefore, we have created a new figure (Fig. 4 in the main text) focusing on specificity.

Figure 4: Improved specificity of LbCas12a trans-cleavage with CRISPR-ENHANCE. (a) Single-point mutations (S1-S20) on the target strand of a dsDNA GFP activator. (b) The heat map displays relative fluorescence intensity normalized to wild-type (WT) activator after 3 hours for a pilot study (n=1). (c) Double-point mutations (D1-D19) on the same target dsDNA GFP activator in (a). (d) Superimposed bar graphs indicating fold change in fluorescence of the mutant activators normalized to the corresponding wild-type activator in (c). (e) Single-point mutations (S1-S20) on the target strand of a dsDNA SARS-CoV-2 activator. (f) Superimposed bar graphs indicating fold change in fluorescence of the mutant activators normalized to the corresponding wild-type activator in (e). (g) Double-point mutations (D1-D19) on the same target dsDNA SARS-CoV-2 activator in (e). (h) Superimposed bar graphs indicating fold change in fluorescence of the mutant activators normalized to the corresponding wild-type activator in (g). Error bars in (d), (f), and (h) represent \pm SEM, where n = 6 from 2 separate experiments; two-way ANOVA test two-way ANOVA (n=3, N=2, P > 0.05 (no asterisks), **P < 0.01, ****P < 0.0001). A fold change in specificity was calculated and reported for only statistically significant mutants by taking the ratio of the normalized data for crRNA-WT to crRNA-3'DNA7. The experiments were repeated at least twice with n = 3 per experiment.

Supplementary Figure 14. Effect of single-point and double-point mutations on the target strand of the double-stranded GFP fragment demonstrated as raw fluorescence signals of Figure 4a-d. **(a)** Sequence representation of dsDNA GFP WT and single-point mutants. The heat map of **(b)** the raw fluorescence intensity and **(c)** the normalized fluorescence intensity with respect to the wild-type activator ($n=1$). The fluorescence data in **(b)** and **(c)** were recorded using ClarioStar plate reader after 3 hours of incubation in a pilot experiment. A portion of this data in **(c)** was plotted in a heat map in the Figure 4b. **(d)** Sequence representation of dsDNA GFP WT and double-point mutants. **(e)** Raw fluorescence intensity and **(f)** the normalized fluorescence intensity with respect to the wild-type activator for 19 double-point mutation of target GFP fragments. The fluorescence data in **(e)** and **(f)** were recorded using BioTek Synergy 2 plate reader after 20 minutes, and the error bars represent \pm SEM, where $n = 2$. The same data in **(f)** was plotted as a bar graph with error bars the Figure 4d. The experiments were repeated at least twice with $n = 2$ per experiment.

Supplementary Figure 15. Effect of single-point and double-point mutations on the target strand of the double-stranded DNA SARS-CoV-2 demonstrated as raw fluorescence signals of Figure 4e-h. **(a)** Sequence representation of dsDNA SARS-CoV-2 WT and single-point mutants. **(b)** Raw fluorescence intensity and **(c)** the normalized fluorescence intensity with respect to the wild-type activator, for 20 single-point mutants of the target SARS-CoV-2. **(d)** Sequence representation of dsDNA SARS-CoV-2 WT and double-point mutants. **(e)** Raw fluorescence intensity and **(f)** the normalized fluorescence intensity with respect to the wild-type activator, after 20 minutes for 19 double-point mutants of the target. All fluorescence data were recorded using BioTek Synergy 2 plate reader, and the error bars represent \pm SEM, where $n = 2$. The same data in **(c)** and **(f)** were plotted as bar graph with error bars in the Figure 4f and Figure 4g, respectively. The experiments were repeated at least twice with $n = 2$ per experiment.

R2.2 It would be interesting to present data using non-fully PS modified DNA. For example, 3'PSDNA7 using 1 to 7 PS substitutions

Response to R2.2- We thank the reviewer for the valuable suggestion and for providing the sequences to be tested. We believe that the suggestion can lead us to have a better mechanistic understanding of LbCas12a post cis-cleavage activity. With your suggestions, we have conducted additional experiments on non-fully phosphorothioate of our crGFP+3'DNA7 with 1 to 6 PS substitutions starting from the 3'-end inwards (the last nucleotide at the 3'-end did not have PS modification due to synthesis issue). The non-fully PS modified DNA versions of crGFP+3'DNA7 has been placed in Fig. 3c,d in the main text.

Additionally, we have added a discussion regarding the non-fully phosphorothioate DNA modifications of the crRNA+3'DNA7 in the main text (see highlighted text below):

We sought to further explore the possibility of this modified crRNA by carrying out experiments on non-fully phosphorothioate of the crGFP+3'DNA7 with 1 to 6 PS substitutions starting from the 3'-end of the extension inwards. We were interested in understanding if the trans-cleavage activity of LbCas12a could be enhanced further by protecting the DNA extension with phosphorothioate modifications. Interestingly, fluorescence-based reporter assays showed that the LbCas12a trans-cleavage activity decreased as more phosphorothioate modifications were added to the extension, with the non-phosphorothioated crRNA+3'DNA7 exhibiting highest fluorescence signal (Fig. 3c,d and Supplementary Fig. 10).

Supplementary Figure 10. Typhoon image (scanned with Amersham Typhoon, GE Healthcare) of the LbCas12a fluorescence-based reporter assay in Figure 2c,d in the main text.

Figure 3: Characterization of CRISPR-ENHANCE with various crRNA modifications and different Cas12a systems. (a) Comparison of trans-cleavage activity between precursor crRNA (pre-crRNA) and mature crRNA (tru-crRNA, where the first Uracil on the 5'-end of the crRNA is cleaved by LbCas12a in the absence of the activator). (b) Comparison of trans-cleavage activity between AT-rich extensions and GC-rich 7-nt DNA 3'-end extensions on the crRNA+3'DNA7. (c) Trans-cleavage activity of LbCas12a with non-fully phosphorothioate (PS) modified crRNA targeting GFP fragment. Sequence representation of 6 non-fully PS extension on the 3'-end of crGFP ranging from 1 to 6 PS. The asterisk symbol (*) signifies the phosphorothioated nucleotide. The graph below the sequence representation shows fold change of the LbCas12a fluorescence-based reporter assay with the activator normalized to the corresponding samples without the activator at t = 30 minutes. (d) kinetics of the LbCas12a fluorescence-based reporter assay in (c). (e) Trans-cleavage activity of different variants of Cas12a. The prefix Lb, As, and Fn stand for Lachnospiraceae bacterium, Acidaminococcus, and Francisella novicida, respectively. Error bars represent \pm SEM, where n = 6 replicates. The experiments were repeated at least twice with n = 3 per experiment.

R2.3 Along the same lines, 2'-DNA and 3'-RNA modifications studies would be a nice addition to this work.

Response to R2.3- We thank the reviewer for the suggestion. However, we believe that these modifications are currently beyond the focus of our manuscript. We hope to test them in the future.

R2.4 Moving forward as a detection tool, assay selectivity and discrimination are critical. Data demonstrating that ENHANCE can discriminate between SARS-CoV, MERS-CoV and SARS-Cov2 for instance, would really strengthen the manuscript.

Response to R2.4- We appreciate the reviewer's suggestion. We agree with the reviewer that assay selectivity and discrimination are critical for our CRISPR-ENHANCE moving forward. Therefore, we have conducted additional experiments to address this point. All the data regarding target specificity against similar pathogens have been presented in figure 7 in the main text of our revised manuscript. We have tested two different types of targets (N-gene plasmid controls purchased from IDT and pathogens' genomic RNA obtained from BEI resources) of SARS-CoV-2 against MERS-CoV, bat-SL-CoVZC45, SARS-CoV Urbani, and HCoV-NL63. There were two pairs of crRNAs (wild-type and ENHANCE versions) programmed to target two separate regions of the SARS-CoV-2 N gene (referred to as N1 and N2). The crRNA targeting the N1 region was designed for inclusivity test where it would recognize SARS-CoV-2, SARS-CoV, and bat-SL-CoVZC45. This guide RNA allowed us to conclude that the target belongs to the SARS-like coronavirus family. The crRNA targeting N2 region was taken from Broughton et al. for exclusivity test. This guide RNA was designed to specifically recognized SARS-CoV-2 against other highly similar organisms in the same family.

We have also added a discussion regarding the above experimental results in the main text as follow (see highlighted text below):

We investigated the specificity of the CRISPR-ENHANCE by testing crRNAs programmed to target SARS-CoV-2 against coronaviruses such as MERS-CoV, SARS-CoV, bat-SL-CoVZC45, and HCoV-NL63. Two guide RNAs were employed to target two different regions of the SARS-CoV-2 N-gene (referred to as N1 and N2 regions). The N1 region of SARS-CoV2 was selected to have ≤ 2 sequence mismatches with SARS-CoV and bat-SL-CoVZC45. This target region was therefore used to recognize if SARS-like coronaviruses strains are detected. The region N2 was selected from Broughton et al. that was specific for SARS-CoV-2 for exclusivity testing (Fig. 7a). We first targeted SARS-CoV-2, MERS-CoV, and bat-SL-CoVZC45 plasmid controls (purchased from IDT) using these two crRNAs. The engineered N1:crCoV2+3'DNA7 and N2:crCoV2+3'DNA7 showed 3-fold and 7.8-fold higher in fluorescence signal compared to the wild-type N1:crCoV2-WT and N2:crCoV2-WT after 10 minutes of incubation, respectively. Notably, the engineered N1:crCoV2+3'DNA7 exhibited lower in fluorescence signal against

MERS-CoV and bat-SL-CoVZC45, demonstrating 74% enhanced specificity towards SARS-CoV-2 (Figs. 7b,c). We next tested the two guides with clinically relevant extracted genomic RNAs of SARS-CoV-2, SARS-CoV Urbani, and HCoV63 (obtained from BEI resources). Both N1:crCoV+3'DNA7 and N2:crCoV+3'DNA7 showed specificity towards SARS-CoV-2 when an RT-LAMP step was applied (Figs. 7d-f). This specificity was due to the fact that RT-LAMP primers sets were specific for SARS-CoV-2 (alignments not shown). Collectively, our CRISPR-ENHANCE system successfully retained the sequence matching fidelity when in complex with LbCas12a with enhanced specificity and significantly higher sensitivity compared to the wild-type crCoV2.

Figure 7. Specificity of CRISPR-ENHANCE for detecting SARS-CoV-2 genomic RNA. (a) Sequence alignment of similar pathogens from the same family with SARS-CoV-2 that were tested in this study. Two crRNAs including their engineered version were designed to target two regions of the N gene N1 and N2 (N1: crCoV and N2: crCoV) where N2 was reported in *Broughton et al.* Sequences were aligned using ClustalOmega^{29,30}, exported in aln file and graphical enhanced in ESPrpt 3.0³¹. (b) crRNA specificity towards SARS-CoV-2 and other highly similar pathogens from the same family. The targets were dsDNA amplified from plasmid controls 2019-nCoV_N_Positive Control, MERS-CoV Control, and SARS-CoV Control (IDT). (c) Detection reaction in (b) scanned by Typhoon (Amersham, GE healthcare). (d) crRNA specificity towards genomic RNA of SARS-CoV-2 and other genomic RNAs of highly similar pathogens from the same family. The targets were genomic RNA obtained from BEI Resources. (e) Lateral flow assay of (d). (f) Detection reaction in (d) scanned by Typhoon (Amersham, GE healthcare). Error bars represent \pm SEM, where $n = 6$ replicates. The experiments were repeated at least twice with $n = 3$ per experiment.

Reviewers' Comments:

Reviewer #1:

Remarks to the Author:

First of all, I think that this manuscript is much better after adding new results.

Generally, I think that the manuscript shares interesting results and insight to the field.

A few minor comments to think about.

1. Figure 4b

I would suggest to keep it same format as 4d, 4f, and 4h.

Seeing Cas12a being promiscuous and showing even enhanced activation with one point mutation sequence than the WT sequence is unfortunate but it is what it is.

Scientifically, it is still meaningful information that Cas12a tolerate one point mutation very well. It should guide readers to know where to use the technology.

2.

'We observed that the 3'DNA7 modifications on crRNAs enhance specificity by up to 8.8-fold across various off-targets when compared to crRNA-WT. Furthermore, based on the statistical analysis, crRNA+3'DNA7 did not significantly reduce the specificity of detection for any mutant tested'

What statistical analysis was done here? There should be many different ways so further explanation can be helpful.

I would be a bit hesitant to advertise that the system shows up to 8.8-fold across various off-targets.

4h doesn't show that dramatic difference.

8.8 fold mentioned in abstract can be misleading.

It's not wrong but something to think about

3. Figure 7 strengthens the manuscript.

What does 1 nM mean here? It will be helpful to add information for reader

What's the standard amount used for the diagnosis? nM? pmole? copy number?

Reviewer #2:

Remarks to the Author:

The authors did a fantastic job addressing all my concerns and was impressed by their rapid turnaround. I have no further concerns and recommend publication.

Enhancement of trans-cleavage activity of Cas12a with engineered crRNA enables amplified nucleic acid detection

REVIEWERS' COMMENTS:

Reviewer #1 (R1):

First of all, I think that this manuscript is much better after adding new results. Generally, I think that the manuscript shares interesting results and insight to the field. A few minor comments to think about.

Response to R1: We thank the reviewer for all their constructive feedback and kind comments.

R1.1 Figure 4b

I would suggest to keep it same format as 4d, 4f, and 4h.

Seeing Cas12a being promiscuous and showing even enhanced activation with one point mutation sequence than the WT sequence is unfortunate but it is what it is.

Scientifically, it is still meaningful information that Cas12a tolerate one point mutation very well. It should guide readers to know where to use the technology.

Response to R1.1: Yes, we agree with the reviewer that Cas12a seems surprisingly promiscuous for single point mutants. Unlike the 4d, 4f, and 4h, the 4b was a pilot experiment (n=1) and therefore we had used a heat map for 4b and bar graphs with error bars for others. With your suggestions, we now performed additional experiments with three replicates and repeated the experiments twice and plotted the 4b in the same format as others.

R1.2 'We observed that the 3'DNA7 modifications on crRNAs enhance specificity by up to 8.8-fold across various off-targets when compared to crRNA-WT. Furthermore, based on the statistical analysis, crRNA+3'DNA7 did not significantly reduce the specificity of detection for any mutant tested'

What statistical analysis was done here? There should be many different ways so further explanation can be helpful.

I would be a bit hesitant to advertise that the system shows up to 8.8-fold across various off-targets.

4h doesn't show that dramatic difference.

8.8 fold mentioned in abstract can be misleading.

It's not wrong but something to think about

Response to R1.2: We have included the statistical analysis of the test in the figure legend. We used a two-way ANOVA test with Dunnett's multiple comparison test using Graphpad Prism to find the significant differences between the wild-type CRISPR vs. ENHANCE for various activators.

Modified Figure 4 legend: '...the statistical analysis was performed using two-way ANOVA test with Dunnett's multiple comparison test and only significant ($p < 0.05$) values were marked with an asterisk (*) indicated as follows: * $p < 0.05$, ** $p < 0.01$, *** $p < 0.001$, and **** $P < 0.0001$ '

Also, we agree that the 8.8-fold in the abstract can be misleading and therefore we have replaced 8.8-fold with the word 'significant'.

Modified abstract: '...and with significant improvement in specificity for target recognition'

R1.3 Figure 7 strengthens the manuscript.

What does 1 nM mean here? It will be helpful to add information for reader

What's the standard amount used for the diagnosis? nM? pmole? copy number?

Response to R1.3: We could not locate 1 nM in Figure 7, however, we found it in the figure 6. As suggested by the reviewer, we have included the either moles or copy numbers in the text and/or figure, and/or figure legends and highlighted in green.

As the standard amount used for diagnosis varies with the stage of the disease, sample type, and the methods used for diagnosis, it would be difficult to add this information in the text. For example, the mean threshold value of SARS-CoV-2 in nasal swabs is around 1.4×10^3 copies/ μL (<https://jamanetwork.com/journals/jama/fullarticle/2762997>). Most CDC recommended qPCR methods <https://www.fda.gov/media/134922/download> uses 100 μL of nasal swabs and can detect as low as 1 copy/ μL . We have tested as low as 3 copies in our assays, however, 300 copies seemed to be a reasonable cutoff for our assays (See Fig. 6c, SI Fig. 29). While ENHANCE's sensitivity was lower than a qPCR but we are within the range of standard diagnosis amounts.

As editors pointed out, we have removed any claims for diagnosis from the manuscript.

Reviewer #2 (Remarks to the Author):

The authors did a fantastic job addressing all my concerns and was impressed by their rapid turnaround. I have no further concerns and recommend publication.

We thank the reviewer for all the comments and for recommending our manuscript for publication.